# A supramolecular metalloenzyme possessing robust oxidase-mimetic catalytic function

Shichao Xu[1], Haifeng Wu[1], Siyuan Liu[1], Peidong Du[1], Hui Wang[2] ✉, Haijun Yang[3], Wenjie Xu[4], Shuangming Chen [4], Li Song [4], Jikun Li[5], Xinghua Shi [2] & Zhen-Gang Wang [1] ✉

Enzymes fold into unique three-dimensional structures to distribute their reactive amino acid residues, but environmental changes can disrupt their essential folding and lead to irreversible activity loss. The de novo synthesis of enzyme-like active sites is challenging due to the difficulty of replicating the spatial arrangement of functional groups. Here, we present a supramolecular mimetic enzyme formed by self-assembling nucleotides with fluorenylmethyloxycarbonyl (Fmoc)-modified amino acids and copper. This catalyst exhibits catalytic functions akin those of copper cluster-dependent oxidases, and catalytic performance surpasses to date-reported artificial complexes. Our experimental and theoretical results reveal the crucial role of periodic arrangement of amino acid components, enabled by fluorenyl stacking, in forming oxidase-mimetic copper clusters. Nucleotides provide coordination atoms that enhance copper activity by facilitating the formation of a copper-peroxide intermediate. The catalyst shows thermophilic behavior, remaining active up to 95 °C in an aqueous environment. These findings may aid the design of advanced biomimetic catalysts and offer insights into primordial redox enzymes.

Metalloenzymes catalyze reactions through the cooperation between exquisitely-arranged amino acid residues and metal-based cofactors. Protein structure is susceptible to environmental changes, which often cause non-recoverable loss of enzyme activity. Burgeoning research into enzyme-inspired supramolecular self-assembly and catalysis, which entails creating enzyme-like active sites, has achieved chemical reactions with substrates similar to those catalyzed by enzymes[1–4]. Despite some advancements toward synthetic enzyme mimics of proteins[5–12], the de novo creation of enzyme-like metallocluster active sites, which have a strict requirement for the geometric distribution of ligand groups and are capable of catalyzing some of the most remarkable reactions in nature, remains largely unresolved.

Metallocluster-dependent enzymes are capable of catalyzing the most basic oxidation-reduction transformations, which generate both chemical building blocks and energy to fuel metabolic processes[13]. In these enzymes, such as nitrogenases, hydrogenases, methane monooxygenases and catechol oxidases, electronic interactions between metals allow the metallocenters to bind the substrates and shuttle electrons into and out of the active site[13]. In the example of catechol oxidases (a family of copper proteins), three types of Cu (Type 1 Cu [T1Cu], Type 2 Cu [T2Cu] and Type 3 Cu [T3Cu]), with distinct coordination spheres, have been identified in the active sites[14–16]. T2Cu is coordinated in either a square-planar or distorted tetrahedral geometry by two nitrogen atoms from a histidine residue, with the

[1]State Key Laboratory of Organic-Inorganic Composites, Key Lab of Biomedical Materials of Natural Macromolecules (Beijing University of Chemical Technology, Ministry of Education), Beijing Laboratory of Biomedical Materials, College of Materials Science and Engineering, Beijing University of Chemical Technology, Beijing 100029, China. [2]Laboratory of Theoretical and Computational Nanoscience, CAS Center for Excellence in Nanoscience, National Center for Nanoscience and Technology, Beijing 100190, China. [3]Department of Chemistry, Tsinghua University, Beijing 10084, China. [4]National Synchrotron Radiation Laboratory, CAS Center for Excellence in Nanoscience, University of Science and Technology of China, Hefei 230029, China. [5]Institute of Chemistry, Chinese Academy of Sciences (ICCAS), Beijing 100190, China. ✉e-mail: wangh@nanoctr.cn; wangzg@mail.buct.edu.cn

remaining site ligated by water. T3Cu contains a dicopper core, in which each copper atom is coordinated by three histidine residues and bridged by a hydroxyl ion. T2Cu and T3Cu are organized into a tri-nuclear copper cluster (TNC)[14,17]. Moreover, there is coordination unsaturation for the copper cluster of the natural enzymes, despite the abundance of ligands (e.g., the side chain imidazole and the backbone carbonyl groups) available from the protein chains. These structural characteristics facilitate oxygen binding and activation at the copper site, promoting the catalytic oxidation. Several groups have attempted to create catechol oxidase-mimicking catalysts[18–22]. However, the structural complexity of the coordination spheres of the synthetic copper complexes was far from those of natural copper centers, leading to considerably lower activity than natural enzymes (e.g., laccase).

Native enzymes rely on well-defined tertiary structures, which are formed through noncovalent interactions in the protein chain, to organize essential functional groups in a pocket where the active sites are created. We were inspired to arrange ligands in an ordered manner, via designed molecular self-assembly (or folding), for clustering metals. The noncovalent interactions among the molecular building block may allow the self-assemblies to have a great structural flexibility, similar to native enzymes, which facilitates the access of the molecular substrates to the active sites inside the supramolecular entity. This approach presents the challenge of producing coordination spheres through the noncovalent association of small molecules. Successful generation of such self-assembled functional groups may shed light on the poorly understood mechanisms by which the mixture of simple molecules and metals evolved into metallocluster-dependent enzymes.

Here, we describe the self-assembly of $Cu^{2+}$ with nucleotides and fluorenylmethyloxycarbonyl (Fmoc)-modified amino acid components to construct catechol oxidase-like copper-cluster active sites (Fig. 1). The fluorenyl moiety with an ortho-fused tricyclic structure

may stack in a directional manner, which allows for the ordered arrangement of ligand groups from the side chain or backbones of the amino acids, for creating coordinatively unsaturated copper centers. Nucleotides can provide nucleobase or phosphate ligands[23,24]. It is expected that these two components can self-assemble to create coordination spheres of $Cu^{2+}$ with a chemical environment similar to the enzymatic copper site. Here, we demonstrate the generation of hybrid self-assemblies that exhibit synergistic catechol oxidase-mimicking activities that were significantly greater than previously reported synthetic catechol oxidase mimics. Moreover, the catalysts showed thermophilic activities, suggesting a robust catalytic effect.

## Results
### Structures of the assemblies
We first used guanosine monophosphate (GMP) and Nα-9-fluorenylmethoxycarbonyl-lysine (Fmoc-K; Supplementary Fig. 1) to self-assemble with $Cu^{2+}$. SEM, AFM and TEM imaging revealed that GMP/$Cu^{2+}$ or Fmoc-K/$Cu^{2+}$ self-assembled into nanoscale fibers, while self-assembly of GMP with Fmoc-K and $Cu^{2+}$ produced irregular nanosheets with lateral dimensions of less than 500 nm and heights of less than 20 nm (Fig. 2a, b, Supplementary Fig. 2 and Supplementary Fig. 3). The circular dichroism (CD) spectrum of GMP shows positive bands at 280 nm, a negative band at ca. 250 nm and a positive long-wavelength tail at 304 nm. The bisignate CD signal is characteristic of a G-quartet maintained by π stacking and hydrogen bonding (Supplementary Fig. 4)[25–27]. The CD spectrum of Fmoc-K/$Cu^{2+}$ fibers displays a negative peak at ca. 212 nm and an absorbance between 240 nm and 320 nm, which can be ascribed to backbone hydrogen bonding (e.g., a mixed β-sheet and disordered structure) and transitions from fluor-enyl−fluorenyl interactions. The CD spectra of the self-assembled Fmoc-K/GMP/$Cu^{2+}$ at different $Cu^{2+}$ concentrations exhibit the spectral features of GMP/$Cu^{2+}$ and Fmoc-K/$Cu^{2+}$, but not simply an overlap of their respective spectra (Supplementary Fig. 4). Fmoc-K/$Cu^{2+}$ exhibits a

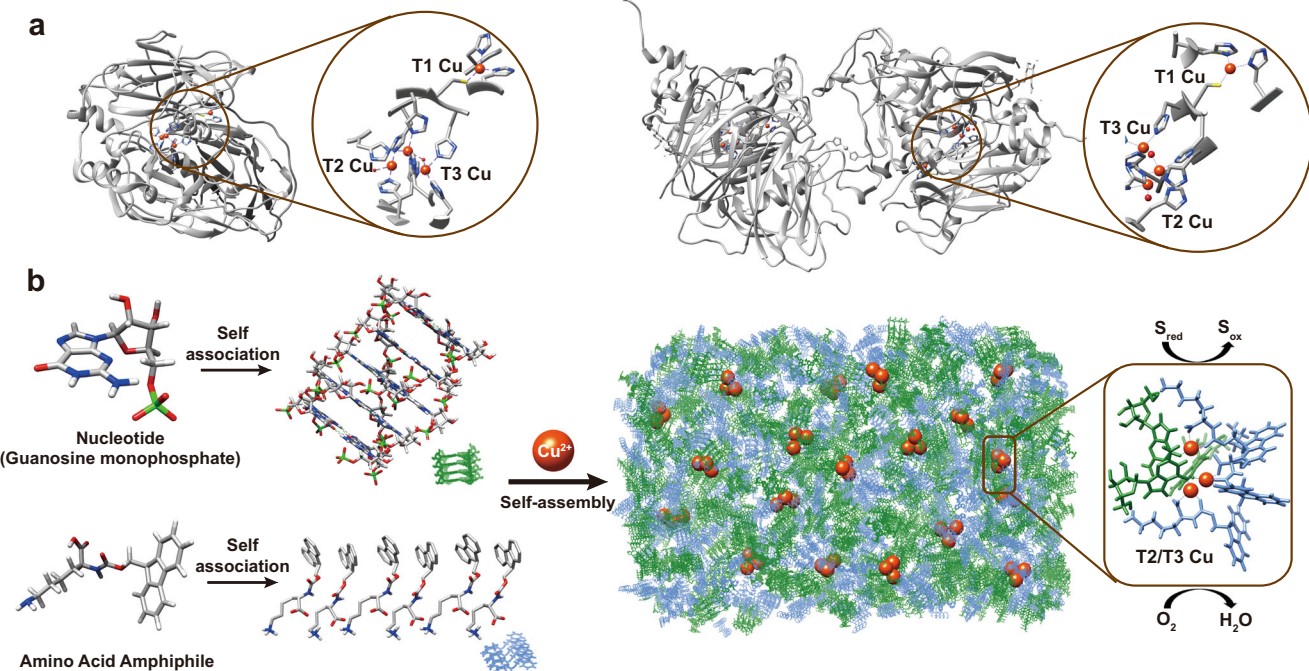

**Fig. 1 | Design of copper-dependent supramolecular catalysts inspired by natural catechol oxidases. a** Structures of *Trametes versicolor* laccase (PDB: 1GYC) and ascorbic acid oxidase (PDB: 1AOZ), showing the copper cluster and distributed residues in the active sites. **b** Schematic self-assembly of nanoaggregates comprising nucleotides (e.g., guanosine monophosphate, GMP), amphiphilic amino

acids (e.g., Fmoc-Lysine-OH, Fmoc-K) and $Cu^{2+}$ ions, that form oxidase-mimetic copper cluster active sites and catalytic activities. Atom colors in the chemical structures: Nitrogen, blue; oxygen, red; phosphorous, green; carbon, dim gray; hydrogen, light gray.

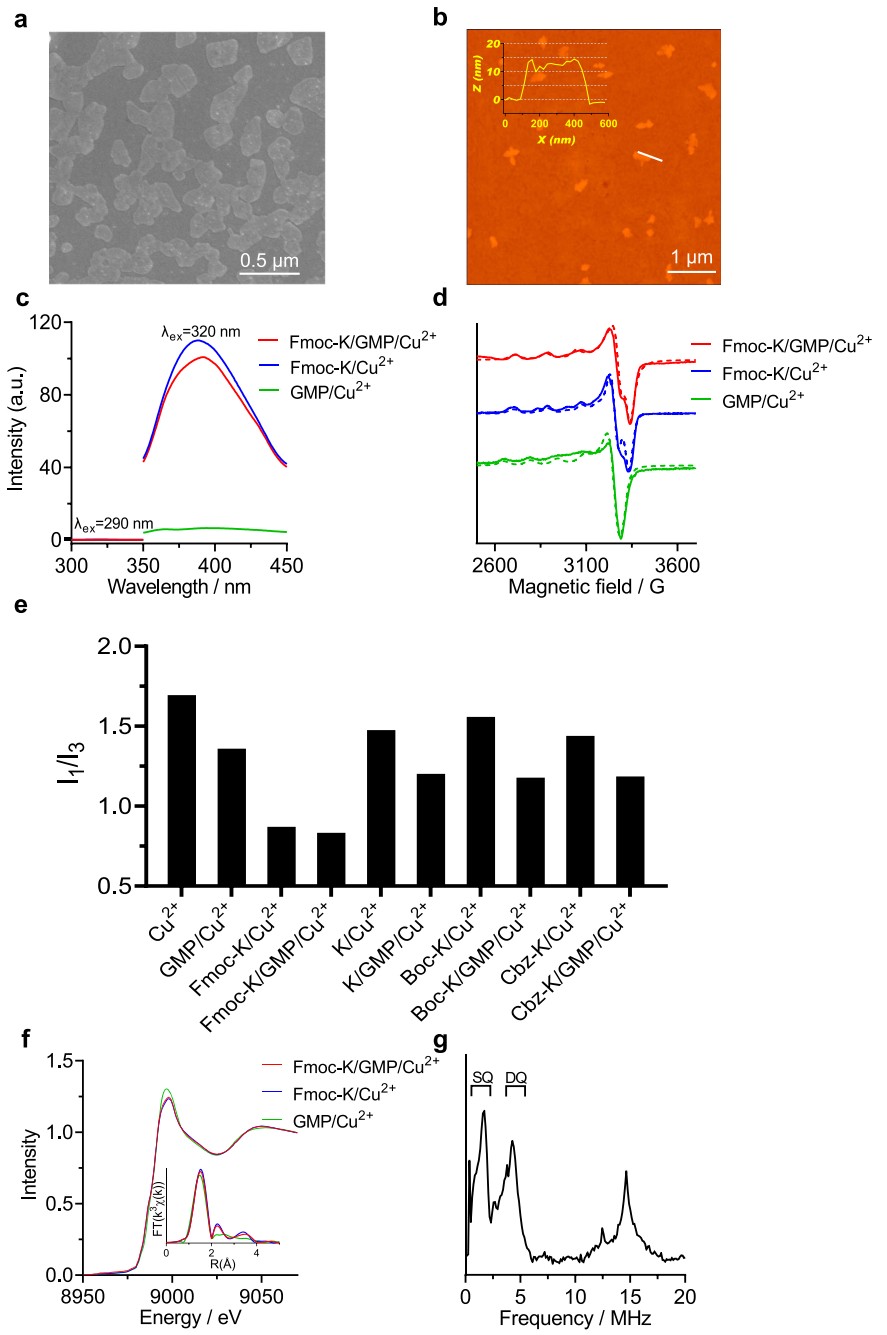

**Fig. 2 | Structural analysis of the Cu²⁺-containing complexes. a** SEM and **b** AFM images of the self-assembled Fmoc-K/GMP/Cu²⁺ complex. Inset is the cross-section analysis of the sample. [GMP]: 10 mM. [Fmoc-K]: 5 mM. [Cu²⁺]: 5 µM. $n$ = 3 independent experiments. **c** Fluorescence emission spectra of the GMP/Cu²⁺, Fmoc-K/Cu²⁺ and Fmoc-K/GMP/Cu²⁺ complexes. [GMP]: 10 mM. [Fmoc-K]: 5 mM. [Cu²⁺] = 5 µM. **d** Experimental (solid) and simulated (dotted) EPR spectra of Fmoc-K/GMP/Cu²⁺, Fmoc-K/Cu²⁺ and GMP/Cu²⁺. [GMP]: 10 mM. [Fmoc-K]: 5 mM. [Cu²⁺] = 100 µM. **e** $I_1/I_3$ values of the mixtures of pyrene and Cu²⁺-containing complexes. [Pyrene] = 5 µM, [Cu²⁺] = 5 µM. [GMP] = 10 mM, [Fmoc-K] = 5 mM, [K] = 5 mM, [Boc-K] = 5 mM, [Cbz-K] = 5 mM. **f** K-edge X-ray absorption spectra and Fourier transforms of the $k^3$-weighted EXAFS modulations for Fmoc-K/GMP/Cu²⁺ (red), Fmoc-K/Cu²⁺ (blue) and GMP/Cu²⁺(green). [GMP]: 10 mM. [Fmoc-K]: 5 mM. [Cu²⁺]: 200 µM. **g** Three-pulse ESEEM spectrum at 3436 G of Fmoc-K/GMP/Cu²⁺. τ = 200 ns. $k$ = 1.81, $\eta$ = 0.43. [Cu²⁺] = 250 µM. [Fmoc-K] = 5 mM, [GMP] = 10 mM.

fluorescence emission at 384 nm, which is characteristic of a parallel arrangement of fluorenyl rings[28–31], along with a very slight emission at 315 nm, which is characteristic of free Fmoc groups without self-assembly (Fig. 2c)[28,29]. No fluorescence emission of GMP was found. A small decrease in the emission intensity of Fmoc-K/Cu²⁺ was observed after assembly with GMP, indicating that GMP slightly interfered with the aggregation of Fmoc-K. These results suggest the successful assembly of GMP, Fmoc-K and Cu²⁺, retaining the self-associating structures of GMP and Fmoc-K.

We evaluated the ground-state electronic states of the Cu²⁺ by low-temperature X-band continuous-wave electron paramagnetic resonance (EPR; Fig. 2d and Supplementary Fig. 5). All the simulated EPR spectra were generated using MATLAB toolbox EasySpin to obtain the g (g∥ and g⊥) and A values[32], which reflect the coordination environments and the unpaired copper electrons in the $d_{x^2-y^2}$ orbital pointing directly at the ligands[33–35]. In particular, the g-shift Δg (≈ g - g_e) (g_e, free electron g factor, ~2.0023) is frequently used to indicate the strength of the crystal field. As shown in Supplementary

Table 1, 2 and 3, the complexes show similar $\Delta g_\perp$ ($\approx g_\perp - g_e$). The Fmoc-K/Cu$^{2+}$ complex exhibited two sets of characteristic EPR parameters ($g_{//}$, 2.300, $A_{//}$, $168 \times 10^{-4}$ cm$^{-1}$; $g_{//}$, 2.250, $A_{//}$, $187 \times 10^{-4}$ cm$^{-1}$), attributed to two Cu (II) species with distinct coordination environments. The EPR spectra of Fmoc-K/GMP/Cu$^{2+}$ show only a single EPR hyperfine feature ($g_{//}$, 2.269, $A_{//}$, $175 \times 10^{-4}$ cm$^{-1}$), the parameters of which were different from those of the Fmoc-K/Cu$^{2+}$ and GMP/Cu$^{2+}$ complexes ($g_{//}$, 2.350, $A_{//}$, $158 \times 10^{-4}$ cm$^{-1}$), indicating a significant change in the ligand environments of Cu$^{2+}$, upon Fmoc-K and GMP self-assembly. This result may also reflect the coordination of Cu$^{2+}$ to both Fmoc-K and GMP components. The $g_{//}$ and $A_{//}$ values of Fmoc-K/Cu$^{2+}$ and GMP/Fmoc-K/Cu$^{2+}$ are consistent with the formation of a paramagnetic T2-Cu site ($g_{//}$, 2.22-2.30, $A_{//}$,158-200 $\times 10^{-4}$ cm$^{-1}$)[15,16,36,37]. To further understand the structure of the Cu$^{2+}$ site, we obtained XAS (X-ray absorption spectroscopy) spectra of these copper complexes at the Cu-K edge (XAFS), which exhibited a intense transition at ~8986 eV and a very weak 1s → 3d pre-edge transition at ~8979 eV, indicating the oxidation state of copper in these complexes[38]. Very similar 1s → 3d transition peak between Fmoc-K/Cu$^{2+}$ and Fmoc-K/GMP/Cu$^{2+}$ complexes indicate they have similar coordination structures of copper[39]. The Fourier transform (FT) data of extended X-ray absorption fine structure (EXAFS) (Fig. 2f) show two Cu-Cu distances in both Fmoc-K/Cu$^{2+}$ (-2.59 Å, -3.75 Å) and GMP/Fmoc-K/Cu$^{2+}$ (-2.59 Å, -3.84 Å), the longer distances are similar to the dicopper of T3Cu (-3.43 Å). There were no signals corresponding to the Cu-Cu observed in GMP/Cu$^{2+}$, and its g and A values were closer to those of CuSO$_4$ in water ($g_{//}$, 2.389, $A_{//}$ $134 \times 10^{-4}$ cm$^{-1}$). These spectral results indicate that there were both T2-mimetic single copper (paramagnetic) and T3Cu-mimetic dicopper sites in the Fmoc-K-based self-assemblies.

We further investigated the formation of Cu$^{2+}$ sites by varying the molecular structures of the building blocks (Supplementary Fig. 1), to determine the possible coordination groups. In some cases, we changed the hydrophobic segment of the amphiphilic amino acids from Fmoc to a tert-butoxycarbonyl (Boc) or benzyloxycarbonyl (Cbz) moiety to yield Boc-K or Cbz-K, or removed the hydrophobic segment to yield unmodified lysine (K). Alternatively, we altered the hydrophilic segment from lysine to arginine (R) or histidine (H) to yield Fmoc-R or Fmoc-H (other Fmoc-modified amino acids were not investigated, due to poor water solubility). Histidine is the main residue providing the coordinating groups for natural copper-cluster sites. We found that Boc-K/GMP/Cu$^{2+}$ ($g_{//}$, 2.254, $A_{//}$, $184 \times 10^{-4}$ cm$^{-1}$) or Cbz-K/GMP/Cu$^{2+}$ ($g_{//}$, 2.349, $A_{//}$, $157 \times 10^{-4}$ cm$^{-1}$) exhibited hyperfine splitting patterns and parameters almost identical to Boc-K/Cu$^{2+}$ ($g_{//}$, 2.251, $A_{//}$, $181 \times 10^{-4}$ cm$^{-1}$) or Cbz-K/Cu$^{2+}$ ($g_{//}$, 2.299, $A_{//}$,$161 \times 10^{-4}$ cm$^{-1}$; Supplementary Fig. 6). On the other hand, Cu$^{2+}$ that self-assembled with Fmoc-H/GMP ($g_{//}$, 2.244, $A_{//}$, $175 \times 10^{-4}$ cm$^{-1}$) or Fmoc-R/GMP ($g_{//}$, 2.26, $A_{//}$, $184 \times 10^{-4}$ cm$^{-1}$) hybrids showed spectral features and hyperfine parameters distinct from those of the individual components (Supplementary Fig. 7 and Supplementary Table 1). The EPR results indicate that self-assembly of Fmoc-modified amino acids with GMP altered the ligand environment of Cu$^{2+}$ in a different manner from Boc- or Cbz-modified residues. The Fmoc fluorenyl group has stronger aromatic stacking and hydrophobic interactions than the Cbz and Boc groups, which lends Fmoc a stronger propensity for intermolecular self-assembly, and may also lead to more effective interactions of the amino acids with GMP. We verified the aromatic stacking of fluorenyl rings in Fmoc-containing amphiphiles by fluorescence emission at 384 nm (Supplementary Fig. 8). The fluorescence of pyrene (the intensity ratio of the first and third vibronic peaks, $I_1/I_3$) was used to probe the polarity of the microenvironments within the self-assembled complexes[40,41]. The $I_1/I_3$ values of pyrene in Fmoc-K/GMP/Cu$^{2+}$ (-0.83) and Fmoc-K/Cu$^{2+}$ (-0.87) were significantly lower than that in water (-1.69) and of other lysine-containing complexes, reflecting the greater hydrophobic association of the Fmoc moiety than Boc or Cbz (Supplementary Fig. 9 and Fig. 2e).

Boc-K/GMP/Cu$^{2+}$ (-2.67 Å, -3.83 Å), Cbz-K/GMP/Cu$^{2+}$ (-2.60 Å, -3.69 Å) and Fmoc-R/GMP/Cu$^{2+}$ (-2.59 Å, -3.64 Å) also show two Cu-Cu distances, while Fmoc-H/GMP/Cu$^{2+}$ only exhibits one Cu-Cu distance (-3.89 Å) (Supplementary Fig. 10), which indicates that T3Cu-mimietic dicopper sites were formed in these self-assemblies.

We changed the base portion of the nucleotide from guanine (GMP) to adenine, cytosine or uracil to yield adenosine monophosphate (AMP), cytidine monophosphate (CMP) or uridine monophosphate (UMP) (for molecular structures, see Supplementary Fig. 1). The N3 position of uracil must deprotonate (pKa 9.9) before it can form a moderate coordination with a metal, while the carbonyl oxygen has a weak affinity for metal[23]. The N7 position of guanine (pKa 2.1), N1 position of adenine (pKa 3.5) and N3 position of cytosine (pKa 4.2) have deprotonated sites for coordination to Cu$^{2+}$, while the oxygen atoms exert a cooperative chelation effect[23]. As shown in Supplementary Fig. 11 and Supplementary Fig. 12, the Fmoc-K/CMP/Cu$^{2+}$ ($g_{//}$, 2.286, $A_{//}$, $175 \times 10^{-4}$ cm$^{-1}$) and Fmoc-K/AMP/Cu$^{2+}$ ($g_{//}$, 2.291, $A_{//}$, $175 \times 10^{-4}$ cm$^{-1}$) exhibit one Cu species with similar hyperfine parameters, although the CMP/Cu$^{2+}$ shows a $^{14}$N superhyperfine pattern in the perpendicular ($g \perp$) region corresponding to four nitrogen nuclei around the Cu$^{2+}$ ion. It is noteworthy that Fmoc-K/UMP/Cu$^{2+}$ shows two Cu species, the g and A values ($g_{//}$, 2.308, $A_{//}$, $167 \times 10^{-4}$ cm$^{-1}$; $g_{//}$, 2.250, $A_{//}$, $184 \times 10^{-4}$ cm$^{-1}$) of which were almost identical to those of Fmoc-K/Cu$^{2+}$, indicating UMP may have a negligible effect on the coordinate environment of Fmoc-K/Cu$^{2+}$. This may be attributed to the weak coordination of the UMP base to Cu$^{2+}$[23], as indicated by the similar hyperfine parameter of UMP/Cu$^{2+}$ to that of CuSO$_4$, compared to GMP/Cu$^{2+}$, CMP/Cu$^{2+}$ and AMP/Cu$^{2+}$. We then changed the monophosphate group to diphosphate, or triphosphate, to yield GDP or GTP, or removed the group to yield guanosine. We observed that the hyperfine parameters of the Fmoc-K/GTP/Cu$^{2+}$ and Fmoc-K/GDP/Cu$^{2+}$ complexes were almost the same as those of CuSO$_4$ (Supplementary Table 1 and Supplementary Fig. 13). The additional phosphate groups in GDP and GTP may both localize Cu$^{2+}$ within the nucleotide components, due to multi-dentate complexation and electrostatic attraction between the phosphate ions and Cu$^{2+}$[24], and reduce the cooperation effect of the nucleobase and amino acid components. It is worth noting that our observation of EPR signals within Fmoc-K/guanosine/Cu$^{2+}$ ($g_{//}$, 2.252, $A_{//}$, $181 \times 10^{-4}$ cm$^{-1}$), which were similar to Fmoc-K/GMP/Cu$^{2+}$, indicates that the effective coordination groups of the nucleotides were from the base to form T2Cu, rather than the phosphate (Supplementary Fig. 14). However, the phosphate groups can improve the solubility of the nucleobases, enhance the association between the nucleobases and amphiphilic acids and buffer the pH of the reaction mixture. The EPR results show that the $\Delta g_{//}$ ($g_{//} - g_e$) (e.g., -0.348 for GMP/Cu$^{2+}$, -0.379 for Fmoc-K/GTP/Cu$^{2+}$ and 0.389 for Fmoc-K/GDP/Cu$^{2+}$) of the Cu$^{2+}$ complexes containing GDP or GTP, or single nucleotide components, were higher than other complexes containing T2Cu-mimietic sites. The higher $\Delta g$ values indicate a lower crystal-field strength[42] as a result of the phosphate groups replacing the π-acid type ligands, such as histidines in copper enzymes and nucleobases in our complexes, as the ligand coordinating to Cu$^{2+}$[16], and may also indicate that phosphate-Cu$^{2+}$ binding was not dominant in the Fmoc-modified amino acids/monophosphate nucleotides (e.g., GMP) hybrids. Cu-Cu distances of <4.0 Å were also found in Fmoc-K/UMP/Cu$^{2+}$ (Supplementary Fig. 15).

To further characterize the Cu$^{2+}$ coordination, we obtained three-pulse electron spin echo envelope modulation (3P-ESEEM) spectra at a magnetic field that corresponds to the $g_\perp$ region of the continuous-wave EPR signal at X-band frequencies (~9.5 GHz)[43–45]. The spectrum for Fmoc-K/Cu$^{2+}$ (Supplementary Fig. 16a) had only one peak at twice the Larmor frequency (ca. 1.95 MHz), indicating a significantly weak hyperfine coupling between Cu$^{2+}$ and a $^{14}$N from the side chain amino group or backbone amide. The GMP/Cu$^{2+}$ spectrum (Supplementary Fig. 16b) showed three peaks (0.43 MHz, 0.92 MHz, and 1.58 MHz) in

the range of 0–2 MHz, the sum of the former two lower frequencies was close to the highest one, indicating the nuclear quadrupole interactions of a weakly coupled $^{14}$N. The three peaks in the ESEEM spectrum are typical of the remote, moderately-coupled $^{14}$N from a guanine ring bound to Cu$^{2+}$. It is evident that the $^{14}$N-Cu$^{2+}$ coupling in Fmoc-K/Cu$^{2+}$ was much weaker than that in GMP/Cu$^{2+}$, indicating a greater distance between $^{14}$N and Cu$^{2+}$ in the former complex. It is noteworthy that ESEEM is well suited for measuring weak hyperfine couplings, so the direct coordination of N7 of guanine to Cu$^{2+}$ cannot be shown by the ESEEM spectra. The spectrum of Fmoc-K/GMP/Cu$^{2+}$ (Fig. 2g) has two sharp peaks (-0.37 MHz and -1.59 MHz) and one broad shoulder peak (-1.10 MHz) that arise from the single quantum (SQ) transition of $^{14}$N, and a broad feature around 4.00 MHz, due to the double quantum (DQ) transition of the $^{14}$N[45,46]. The ESEEM spectrum of Fmoc-K/GMP/Cu$^{2+}$ was different from that of Fmoc-K/Cu$^{2+}$ or GMP/Cu$^{2+}$, and not a superposition of the two, indicating the contribution by both component ligands to the $^{14}$N coupling to Cu$^{2+}$. Compared with Fmoc-K/Cu$^{2+}$, the ESEEM spectrum of Fmoc-H/Cu$^{2+}$ (Supplementary Fig. 17a) or Fmoc-R/Cu$^{2+}$ (Supplementary Fig. 17b) shows four peaks typical of a weakly coupled $^{14}$N, which indicates greater contribution of imidazole of Fmoc-H and guanidino of Fmoc-R to $^{14}$N-Cu coupling than that of Fmoc-K. Similarly, Fmoc-K/Cu$^{2+}$ (Supplementary Fig. 16a), Cbz-K/Cu$^{2+}$ (Supplementary Fig. 17c) shows only one peak at twice the Larmor frequency in the ESEEM spectrum, again indicating a significantly weak coupled $^{14}$N from Cu$^{2+}$. When these amphiphilic amino acids assembled with GMP, the ESEEM spectra were different from those of each component/Cu$^{2+}$ (Supplementary Fig. 18). All simulation parameters (e$^2$qQ/h and η) are presented in Supplementary Table 4. These $^{14}$N ESEEM results indicate: (i) Fmoc-H and Fmoc-R can provide nitrogen and oxygen as the coordinating atoms, while Fmoc-K can only provide oxygen; (ii) the self-assembly of the heterogeneous components can change the ligand environments of Cu$^{2+}$. The peak at ca. 14.40 MHz for all ESEEM spectra are from the weakly coupled protons that may come from either solvent or the ligand[46].

To gain further insight into the coordination of Cu$^{2+}$ to the guanine ring, we conducted hydrogen nuclear magnetic resonance ($^1$H-NMR) measurements of guanosine/Cu$^{2+}$ in DMSO-d6. Line broadening and shifting are mainly observed on the NMR signals of hydrogen nuclei that are in close proximity to the paramagnetic Cu$^{2+}$ ions. This is due to the interaction between the magnetic field of the unpaired electron of Cu$^{2+}$ and the nearby NMR nucleus, which causes the relaxation time of those nuclei to become shorter[47–49]. As shown in Supplementary Fig. 19, after adding Cu$^{2+}$, a broadening and shifting of the peak at ca. 7.93 ppm was observed, which was attributed to the hydrogen on C8 of guanine ring. This finding confirms the coordination of N7 of guanine to Cu$^{2+}$. The hyperfine constant, A$_{iso}$, is estimated to be 0.18 MHz from $^1$H-NMR shift (Details of the analysis and calculation can be found in methods "Estimation of hyperfine constant from NMR shifts"). No line broadening was observed for the hydrogens of Fmoc-K/Cu$^{2+}$ (Supplementary Fig. 19b), indicating the hydrogens were away from Cu$^{2+}$. A similar line broadening of the guanine ring was also observed for Fmoc-K/guanosine after adding Cu$^{2+}$ (Supplementary Fig. 19a). Based on the results of $^1$H-NMR, cwEPR, and ESEEM, it can be reasoned that the coordination sphere of Cu$^{2+}$ in Fmoc-K/guanosine/Cu$^{2+}$ (or Fmoc-K/GMP/Cu$^{2+}$) was composed of N7 of guanine base and carbonyl/carboxylate of the lysine moiety.

## Structural modeling

We performed density functional theory (DFT) calculations to understand the coordination of Cu$^{2+}$ to Fmoc-K or GMP. The model complexes at the single-molecular level are presented in Supplementary Fig. 20. The NH$_3^+$ side chain of Fmoc-K, with a pKa of 10.79, poorly coordinates to Cu$^{2+}$. Thus, Cu$^{2+}$ is mainly coordinated to the carbonyl oxygen atoms (complex $i$ model). Considering that the phosphate groups did not contribute to the formation of the copper center in

Fmoc-K/GMP hybrids, Cu$^{2+}$ may form the complexes with GMP in two ways (complex ii and/or iii; the phosphate is not shown). In model complex ii, Cu$^{2+}$ is coordinated to the N3 atom and 2′-OH of the ribose moiety, while in complex iii, Cu$^{2+}$ is coordinated to the N7 and O6 atoms. The 2′-OH is absent in the deoxyguanosine monophosphate (dGMP), while the EPR parameters (g$_{//}$, 2.274, A$_{//}$, 173 × 10$^{-4}$ cm$^{-1}$) of the Fmoc-K/$d$GMP/Cu$^{2+}$ complex (Supplementary Fig. 21), reflecting the strength of the crystal field and the coordination environments, and the dicopper distances (2.57 and 3.63 Å, Supplementary Fig. 22), were almost identical to those of Fmoc-K/GMP/Cu$^{2+}$. These results exclude the participation of the ribose (complex ii). Complexes i and iii constitute the basis of the construction of theoretical T2 and T3Cu models.

To reveal the Cu$^{2+}$ center in the supramolecular materials, we investigated the self-assembled structures of Fmoc-K, by resolving the structures of nanoscale Fmoc-K crystals (Supplementary Fig. 23) that were formed after incubating an Fmoc-K aqueous solution for ca. six months. The typical selective-area electron diffraction (SAED) pattern recorded perpendicularly to the surfaces shows a single-crystalline nature of the self-assembled Fmoc-K structures (Fig. 3a and Supplementary Fig. 24). We then performed a first-principles calculation to elucidate the packing structure of the Fmoc-K molecules (Fig. 3b). The simulation revealed that Fmoc-K adopted a triclinic unit cell with the $P$1 space group and cell lengths, $a = 4.593(8)$ Å, $b = 7.281(8)$ Å, $c = 24.841(2)$ Å, and angles, $\alpha = 96.1015°$, $\beta = 96.5859°$, $\gamma = 108.7390°$. The diffraction peaks in the simulated XRD pattern (Supplementary Fig. 25) using this single-crystal data matched those of the SAED pattern (Fig. 3a and Supplementary Fig. 25) and low-temperature (120 K) powder XRD pattern (Supplementary Fig. 26). The Fmoc-K molecules assembled through π-π stacking, electrostatic interactions and hydrogen bonding. The Cu$^{2+}$ can coordinate to Fmoc-K molecules along facet (0 0 1) (Supplementary Fig. 27) or (1 0 0) (Fig. 3c, i) to form a single-copper species with a coordination environment of distinct configuration and crystal-field strength, as observed by EPR (Supplementary Fig. 5). In the Fmoc-K aggregate, the neighboring copper sites along facet (0 0 1) or (1 0 0) (Fig. 3c, ii) can form a dicopper site (Supplementary Fig. 28), corresponding to the observed Cu-Cu distance of 2.59 Å or 3.73 Å (EXAFS, Fig. 2f). It is noteworthy that, in both aggregates, the dicopper of 3.73 Å is expected to be more catalytically active than that of 2.59 Å (also observed in CuO), in which the copper had a lower binding affinity to O$_2$. The GMP that assembled with Fmoc-K slightly disturbed the fluorenyl-fluorenyl stacking (as observed from the fluorescence emission of Fmoc-K, Fig. 2c) leading to the coordination of Cu$^{2+}$ to both GMP and Fmoc-K (Supplementary Fig. 29), and the formation of single-copper sites with a similar crystal field (as revealed by EPR, Fig. 2d). The coordination sphere of the single-copper site in GMP/Fmoc-K was composed of the N7 and O6 atoms from the guanosine moiety and the two carbonyl oxygen atoms from Fmoc-K (Fig. 3d, i). The dicopper sites (Fig. 3d, ii) formed in the Fmoc-K/GMP aggregate were similar to those in the Fmoc-K aggregate, as revealed by EXAFS (Fig. 2f), and had a distance of 3.76 Å. The ligands distributed around the single copper (T2) or dicopper (T3) sites can provide potential binding sites for Cu$^{2+}$ to form additional catalytic centers, which may result in proximal copper arrangement, cooperating like a multinuclear copper site. No difference was observed in the CD (Supplementary Fig. 4) and fluorescence spectra (Supplementary Fig. 30) of the Fmoc-K/GMP hybrid in the presence or absence of Cu$^{2+}$, indicating that the self-assembled structures were not affected by adding Cu$^{2+}$. This suggests that the Fmoc-K or Fmoc-K/GMP self-assembly creates the framework of the coordination sphere for the copper sites. The nucleobase moiety that coordinates to Cu$^{2+}$ may facilitate the proton transfer to the copper center during catalysis[14]. The Fmoc-H packing model was built with crystallized Fmoc as the reference (Supplementary Fig. 31), since the aggregation of these two amphiphilic amino acids can both be dominated by π-π stacking and hydrogen bonding interactions. However, Cu$^{2+}$ coordination occurred on the facet (0 0 1), where flexible imidazole

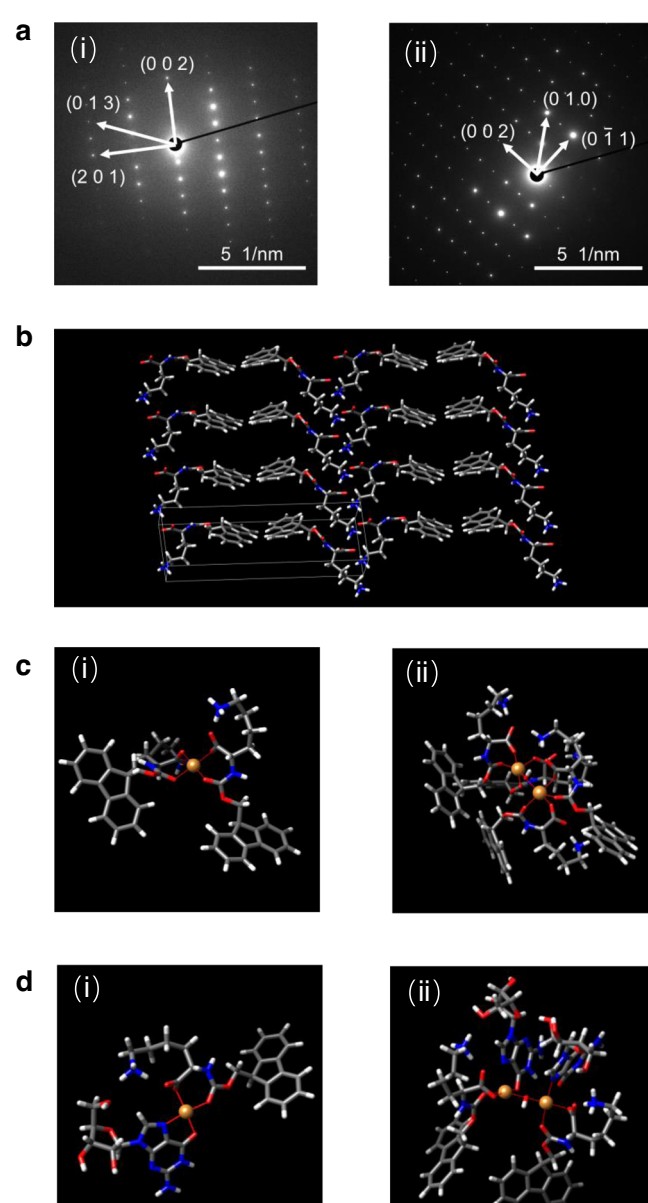

**Fig. 3 | Structural model of the Fmoc-K crystal and Cu²⁺-containing complexes.** **a** SAED patterns at different areas of the Fmoc-K crystal structure (0.1 M). **b** Theoretical model of the molecular packing in the Fmoc-K crystal. Each unit outlined in white contains two Fmoc-K molecules linked through π-stacking interactions. **c** Theoretical model of single copper (i) and dicopper (ii) (facet (1 0 0)) in the Fmoc-K self-assembly, based on the model in **b**. **d** Theoretical model of single copper (i) and dicopper (ii) in Fmoc-K/GMP self-assembly. N, O, C, H and Cu atoms are indicated in blue, red, dim gray, light gray and orange, respectively.

and carbonyl groups are distributed around the Cu²⁺ (Supplementary Fig. 32), leading to a greater coordination number of ligands to copper than that in Fmoc-K/Cu²⁺ (Supplementary Fig. 28). The side chain guanidinium of arginine also has coordination capability[50,51]. It can be concluded that the coordination of Cu²⁺ in Fmoc-R is similar to that in Fmoc-H aggregates.

## Structure-activity relationship

We examined the catalytic activity of the self-assembled Cu²⁺ complexes with dissolved $O_2$ and 2,4-dichlorophenol (2,4-DCP) as the substrates. The 2,4-DCP was oxidized into semiquinone radicals, which then reacted with 4-aminoantipyrine (4-AP) to produce a red

adduct, with a maximum absorbance at 510 nm (Supplementary Fig. 33). The time-dependent-absorbance changes at 510 nm were recorded for the different Cu²⁺ complexes (Fig. 4a and Supplementary Fig. 34); the reaction rate catalyzed by the Fmoc-K/GMP/Cu²⁺ complex was significantly higher than that of Fmoc-K/Cu²⁺ or GMP/Cu²⁺. The initial catalytic velocity ($V_i$) was estimated at different Cu²⁺ concentrations (Fig. 4b); the difference in $V_i$ values was more pronounced at lower Cu²⁺ concentrations. At 5 μM Cu²⁺, the $V_i$ values for the Fmoc-K/GMP/Cu²⁺, GMP/Cu²⁺ and Fmoc-K/Cu²⁺ complexes were 0.525 ± 0.049 μM s⁻¹, 0.003 ± 0.001 μM s⁻¹ and 0.031 ± 0.005 μM s⁻¹, respectively. At 100 μM Cu²⁺, the respective $V_i$ values were 4.675 ± 0.363 μM s⁻¹, 0.062 ± 0.001 μM s⁻¹ and 0.232 ± 0.039 μM s⁻¹. These results reveal a remarkable synergy between the GMP and Fmoc-K components in accelerating the catalysis by Cu²⁺.

To investigate the enzyme kinetics, we plotted $V_i$ against the concentration of the reductants, in the presence of the Fmoc-K/GMP/Cu²⁺ complex or *Trametes versicolor* laccase. The global fitting approach was used to evaluate the turnover rate ($k_{cat}$) and the catalytic efficiency ($k_{cat}/K_m$) per Cu (Supplementary Fig. 35 to Supplementary Fig. 40 and Table 1). We evaluated the amount of Cu in the as-received laccase by inductively coupled plasma mass spectrometry (ICP-MS; 25.1 μg/g). With a chlorophenol (2,4-dichlorophenol [2,4-DCP], 2,4,6-trichlorophenol [2,4,6-TCP], 2,3,5,6-tetrachlorophenol [2,3,5,6-TCP], 3,3′,5,5′-Tetramethylbenzidine [TMB], 3,5-di-tert-butylcatechol [DTBC] or 2,6-dimethoxyphenol [DMP]) as the reducing substrate, the self-assembled complexes all exhibited lower $k_{cat}$ and $k_{cat}/K_m$ values than native laccase. It is notable that the $k^{TMB}_{cat}$ (1.3150 ± 0.0270 s⁻¹) and $k^{DTBC}_{cat}$ (1.1960 ± 0.0370 s⁻¹) of the Fmoc-K/GMP/Cu²⁺ complex reached the same order of magnitude to that of laccase (8.3038 ± 1.7595 s⁻¹ and 5.3164 ± 0.3418 s⁻¹), and $k^{DTBC}_{cat}/K^{DTBC}_m$ (2.643 s⁻¹ mM⁻¹) was over 60% of that of laccase (3.7518 s⁻¹ mM⁻¹). Moreover, the obtained kinetic parameters were markedly greater than previously reported Cu²⁺ complexes in aqueous solution (Supplementary Table 5). The $k_{cat}$ value of the catalyst towards the oxidization of DMP was ca. 2.95 s⁻¹, which is 300-fold higher than that reported for an amyloid short peptide/Cu²⁺ complex, which also showed the EPR splitting features of T2Cu[21]. A markedly lower activity was observed for tyrosinase that only possess T3Cu sites (the copper amount was evaluated by ICP-MS; Supplementary Fig. 41). We conclude that the catalytic reactions may be synergistically accelerated by the T2Cu- and T3Cu-mimicking sites of the self-assembled complexes. The $k_{TMB}$ cat value (1.3150 ± 0.0270 s⁻¹) was almost 2-fold that of a recently reported, single-atom Fe nanoenzyme (0.7084 s⁻¹) and 120-fold that of the commercially available Pt/C (0.0101 s⁻¹)[52]. As shown in Supplementary fig. 34a, in the presence of Fmoc-K/GMP/Cu²⁺ (Cu²⁺,5 μM) and 2,4-DCP (1 mM)/4-AP (1 mM), the absorbance reached to the 4 when the catalyzed reaction proceeded for ca. 10 min. After calculating, one copper can promote oxidation of at least ca. sixty 2,4-DCP molecules, indicating Fmoc-K/GMP/Cu²⁺ is a catalyst. We also calculated the cost of the Fmoc-K/GMP/Cu²⁺ catalyst (ca $12 USD) with laccase (ca. $493 USD; the commercial sources are provided in the Materials and Methods section), with both catalyzing the oxidation of 2,4-DCP with a $V_i$ value of 0.525 μM s⁻¹ in 1 L aqueous solution, revealing that Fmoc-K/GMP/Cu²⁺ is considerably more cost-effective. We also tested the oxidase-mimetic function of a variety of transition metals complexed with the Fmoc-K/GMP hybrid, and found that only Cu²⁺ exhibited catalytic activity (Supplementary Fig. 42a). Moreover, the addition of other transition metals, at equimolar ratios to copper, to Fmoc-K/GMP/Cu²⁺ did not alter the $V_i$ value, revealing a high specificity of the catalytic activity to Cu²⁺ (Supplementary Fig. 42b).

During catalysis by laccase, the clustered Cu²⁺ ions accept electrons from the reducing substrates to transform into Cu⁺ ions, which are then oxidized by $O_2$ to return to Cu²⁺[53]. The observed synergistic activity in the self-assembled Fmoc-K/GMP/Cu²⁺ complex may be due to the enhanced reactivity of Cu with $O_2$. We deaerated the catalytic system containing Fmoc-K/GMP/Cu²⁺ and 2,4-DCP/4-AP, and found that the

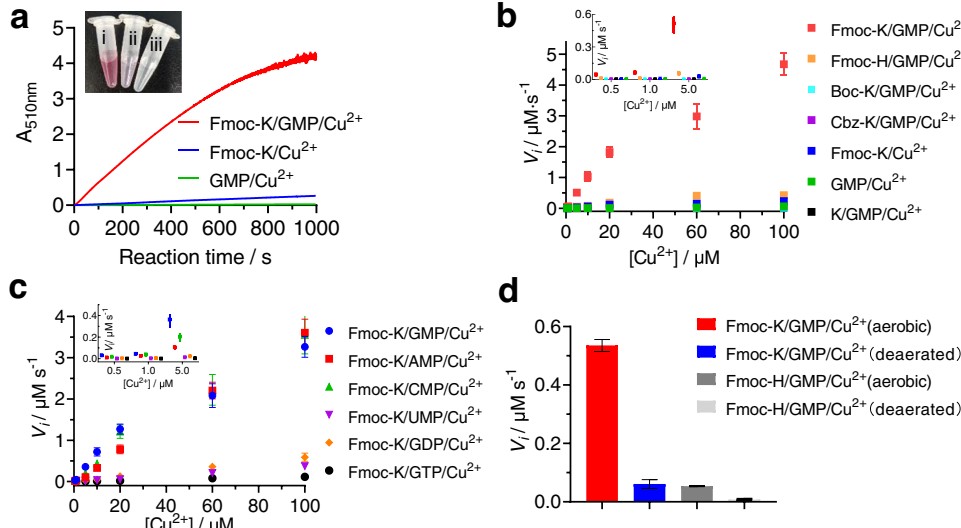

**Fig. 4 | Catalytic activity of the Cu²⁺-containing complexes in 2,4-DCP oxidation, with O₂ as the reducing substrate. a** Time-dependent absorbance changes at 510 nm for the indicated complexes after addition of the substrates. Inset: reaction mixtures at 10 min after initiation. (i) Fmoc-K/GMP/Cu²⁺, (ii) Fmoc-K/Cu²⁺, (iii) GMP/Cu²⁺. [Cu²⁺] = 5 μM. **b, c** The initial catalytic velocities ($V_i$) for the indicated Cu²⁺-containing complexes at various Cu²⁺ concentrations. **d** Initial catalytic velocities for Fmoc-K/GMP/Cu²⁺ and Fmoc-H/GMP/Cu²⁺ under aerobic or deaerated conditions. [Cu²⁺] = 5 μM. The data in panels b, c and d are presented as the mean ± s.d. (*n* = 3). [Fmoc-K] = 5 mM, [Boc-K] = 5 mM, [Cbz-K] = 5 mM, [K] = 5 mM, [Fmoc-H] = 5 mM, [GMP] = 10 mM, [2,4-DCP] = 1 mM, [4-AP] = 1 mM. The data in **b, c** and **d** are presented as the mean ± s.d., with the error bars representing the s.d. and *n* = 3 independent experiments.

formation rate of the red adduct decreased by over 14-fold (Fig. 4d), confirming the key role of O₂ in the reactions. To understand the structure-activity relationship, we examined the effects of the molecular structures of the components on the initial velocity of the reactions. With 2,4-DCP as the substrate, the $V_i$ value of Fmoc-K/GMP/Cu²⁺ was 9.8-fold higher than Fmoc-H/GMP/Cu²⁺ (at 5 μM Cu²⁺; Fig. 4b); when the GMP and Fmoc-K (or Fmoc-H) were 10 mM and 5 mM, respectively, the $k_{cat}/K_m$ value of Fmoc-K/GMP/Cu²⁺ was ca. 30-fold higher than that of

## Table 1 | Apparent kinetic parameters for laccase and Cu-containing artificial complexes with respect to the catalytic oxidation of a variety of reducing substrates

|  | $k_{cat}$ (s⁻¹) | $K_m$ (mM) | $k_{cat}/K_m$ (s⁻¹ mM⁻¹) |
|---|---|---|---|
| Fmoc-K/Cu²⁺ (2,4-DCP) | 0.0232±0.0060 | 5.1200±1.916 | 0.0045 |
| GMP/Cu²⁺ (2,4-DCP) | 0.0010±0.0002 | 2.6660±0.7310 | 0.0004 |
| Fmoc-K/GMP/Cu²⁺ (2,4-DCP) | 0.2236±0.0120 | 0.6962±0.1200 | 0.3211 |
| Fmoc-H/GMP/Cu²⁺ (2,4-DCP) | 0.0751±0.0291 | 6.7900±3.0000 | 0.0111 |
| Laccase (2,4-DCP) | 5.924±0.1266 | 0.2912±0.0314 | 20.6200 |
| Fmoc-K/GMP/Cu²⁺ (2,4,6-TCP) | 1.0960±0.0880 | 0.2494±0.0540 | 4.6908 |
| Laccase (2,4,6-TCP) | 33.7720±1.038 | 0.2677±0.0351 | 128.8610 |
| Fmoc-K/GMP/Cu²⁺ (2,3,5,6-TCP) | 0.0800±0.0052 | 0.0683±0.0047 | 1.1826 |
| Laccase (2,3,5,6-TCP) | 1.767±0.5949 | 0.0440±0.0032 | 41.3633 |
| Fmoc-K/GMP/Cu²⁺ (TMB) | 1.3150±0.0270 | 0.8018±0.1150 | 1.6800 |
| Laccase (TMB) | 8.3038±1.7595 | 0.2374±0.1092 | 48.7215 |
| Fmoc-K/GMP/Cu²⁺ (DTBC) | 1.1960±0.0370 | 0.4530±0.0150 | 2.4247 |
| Laccase (DTBC) | 5.3164±0.3418 | 1.417±0.1370 | 3.1801 |
| Fmoc-K/GMP/Cu²⁺ (DMP) | 2.9500±0.8820 | 19.5700±8.4800 | 0.1507 |
| Laccase (DMP) | 15.6835±0.6139 | 0.0229±0.0029 | 684.2785 |

Fmoc-H/GMP/Cu²⁺ (Supplementary Fig. 43 and Table 1). The activity of Fmoc-R/GMP/Cu²⁺ was slightly higher than that of Fmoc-H/GMP/Cu²⁺, but also significantly lower than that of Fmoc-K/GMP/Cu²⁺ (Supplementary Fig. 44). No activity was observed for the Cu²⁺ complexes containing Boc-K and Cbz-K. All these materials contained dicopper and single copper sites. On the other hand, it is noteworthy that the natural copper sites are coordinatively unsaturated, allowing for O₂ binding to the copper clusters and O₂ reduction, as a result of the well-defined protein folding. As mentioned above, the considerably weaker aromatic stacking and hydrophobic interactions between Boc-K or Cbz-K molecules (Fig. 2e and Supplementary Fig. 45) than those between Fmoc-K molecules may lead to random distribution of the amino acids around Cu²⁺ and the coordination saturation of the metal. Moreover, there is no simultaneous coordination of Cu²⁺ to both Boc-K (or Cbz-K) and GMP, as observed from hyperfine parameters of EPR (Supplementary Table 1), leading to the inefficient catalytic cooperation between these two components. These two factors may lead to the significantly low activity of Boc-K/GMP/Cu²⁺ and Boc-K/GMP/Cu²⁺. The Fmoc-H or Fmoc-R aggregates showed a greater number of ligands around Cu²⁺ than Fmoc-K, leading to the lower activity of Fmoc-H/GMP/Cu²⁺ and Fmoc-R/GMP/Cu²⁺ than Fmoc-K/GMP/Cu²⁺, and Fmoc-H/Cu²⁺ than Fmoc-K/Cu²⁺. The catalytic activity of Fmoc-H/Cu²⁺ and Fmoc-H/GMP/Cu²⁺ was still significantly higher than Boc-K/GMP/Cu²⁺, Cbz-K/GMP/Cu²⁺, Boc-H/GMP/Cu²⁺, and Cbz-H/GMP/Cu²⁺, illustrating the importance of the ordered aromatic packing of the fluorenyl groups to the formation of the highly active copper site.

The $V_i$ value decreased slightly for the Cu²⁺ complexes containing GMP, CMP and AMP (Fig. 4c), following the order of base coordination affinity to Cu²⁺[23], when the Cu²⁺ concentration was below 10 μM. At higher Cu²⁺ concentrations, these complexes showed almost identical activities. In comparison, Fmoc-K/UMP/Cu²⁺ was markedly less active. With varying numbers of phosphate, Fmoc-K/guanosine/Cu²⁺ and Fmoc-K/GMP/Cu²⁺ showed equal activities (Supplementary Fig. 46), while the activity of Fmoc-K/GDP/Cu²⁺ was much lower, and Fmoc-K/GTP/Cu²⁺ was not active (Fig. 4c). The EPR results demonstrate that (i) UMP only had negligible effect on the coordination environments of Cu²⁺ in the Fmoc-K aggregates; (ii) The addition of GDP and GTP to Fmoc-K resulted in the localization of Cu²⁺ to the phosphate

environments because of strong $Cu^{2+}$-phosphate binding. These results indicate that the coordination of $Cu^{2+}$ to both Fmoc-K and nucleotide components is important to the oxidase activity of the $Cu^{2+}$.

Despite their different activities, Fmoc-K/GMP/$Cu^{2+}$ and Fmoc-H/GMP/$Cu^{2+}$ had comparable apparent activation energy (71.708 kJ mol$^{-1}$ and 72.032 kJ mol$^{-1}$), as calculated using the Arrhenius equation (Supplementary Fig. 47)[54–56], indicating that the reactions catalyzed by these two complexes had similar intermediate species and reaction paths. The other less-active or non-active complexes, including Fmoc-K/$Cu^{2+}$ (75.566 kJ mol$^{-1}$), GMP/$Cu^{2+}$ (82.217 kJ mol$^{-1}$), Fmoc-H/$Cu^{2+}$ (125.566 kJ mol$^{-1}$), Fmoc-K/GTP/$Cu^{2+}$ (78.614 kJ mol$^{-1}$), Fmoc-K/UMP/$Cu^{2+}$ (80.513 kJ mol$^{-1}$) and Cbz-K/GMP/$Cu^{2+}$ (77.254 kJ mol$^{-1}$), exhibited higher $E_a$ values.

The inhibitory effect of the imidazole on the binding of the exogenous ligand was further revealed by adding $NaN_3$ to the catalytic system. $NaN_3$ has previously been used as an anion inhibitor of multicopper oxidases (e.g., laccase, ascorbate oxidase), that binds T2Cu and bridges T2Cu to T3Cu[57,58]. Excessive $NaN_3$ almost fully deactivated the native laccase, but unexpectedly, increased the activity of Fmoc-K/GMP/$Cu^{2+}$ (Fmoc-K 5 mM, Fmoc-K/GMP: 1/2) by ca. 25% (Supplementary Fig. 48a). The absorbance in the range of 350 nm–450 nm increased (Supplementary Fig. 48b), which can be attributed to the $N_3^-{\rightarrow}Cu^{2+}$ charge transition[59]. When the concentrations of Fmoc-K and GMP were decreased, the activity enhancement (Supplementary Fig. 48c), as well as absorbance change upon addition of $N_3^-$ (Supplementary Fig. 48d), were more pronounced. We also found that $N_3^-$ can enhance the activity of Fmoc-K/$Cu^{2+}$, but less significantly than GMP (Supplementary Fig. 49). We conclude that the $N_3^-$-induced enhancement of Fmoc-K/GMP/$Cu^{2+}$ activity can be ascribed to the coordination of a greater quantity of $Cu^{2+}$ to both ligands, Fmoc-K and GMP (or $N_3^-$). In contrast to Fmoc-K/GMP/$Cu^{2+}$, $N_3^-$ deactivated both Fmoc-H/GMP/$Cu^{2+}$ and Fmoc-H/$Cu^{2+}$, which was more evident at lower Fmoc-H concentrations (Supplementary Fig. 50). The UV-vis spectra show only slight $N_3^-{\rightarrow}Cu^{2+}$ charge transitions, compared to Fmoc-K/GMP/$Cu^{2+}$ with $N_3^-$ (Supplementary Fig. 51). The mechanism of $N_3^-$-induced deactivation is probably due to the hydrogen bonding of $N_3^-$ to the imidazole of the neighboring Fmoc-H, which leads to an enhancement of the basicity of the imidazole and its coordination to copper, and decreased $N_3^-$ coordination capability. $N_3^-$ decreased the intensity of the fluorescence emission of stacked Fmoc-His at 384 nm, while not affecting that of Fmoc-K (Supplementary Fig. 52). Meanwhile, the multivalent, strong coordination of Fmoc-His to $Cu^{2+}$ also disfavors the $Cu^{2+}$-$N_3^-$ binding.

## Catalytic mechanism

To further understand the catalytic mechanism of the enzyme mimic, we performed density functional theory (DFT) to simulate the adsorption of $O_2$ to the dicopper site (Fig. 5a), which represents the minimal unit of copper cluster in the aggregate. Successful $O_2$ adsorption is a key event leading to the subsequent catalytic reactions. The O-O bridged the copper to form a peroxide species. Figure 5b shows the possible pathway for the catalyzed $O_2$ reduction, with the catalytic mechanism of laccase as a refs. 14,60–62. The reaction begins when the reductant (e.g. 2.4-dichorophenol, 2,4,6-trichlorophenol, 3,3′,5,5′-tetramethylbenzidine or 3,5-ditert-butyl catechol) donates two electrons to reduce the $Cu^{2+}$ to $Cu^+$ (State i to ii). Next, the reduced dicopper reacts with $O_2$ to generate a "peroxide intermediate" (PI; State iii), where the two copper atoms are oxidized and have reduced dioxygen by two electrons. The peroxide intermediate decays to the native intermediate (NI; State iv), by accepting two electrons from other reductants, transferring two protons and cleaving the O-O bond. The NI can decay to the resting oxidized (RO) form, which is reduced to start a new catalytic cycle. In short, the dicopper site couples four $1e^-$ oxidations of the substrates to the $4e^-$ reduction of $O_2$ to water. We also modeled the adsorption of $O_2$ to

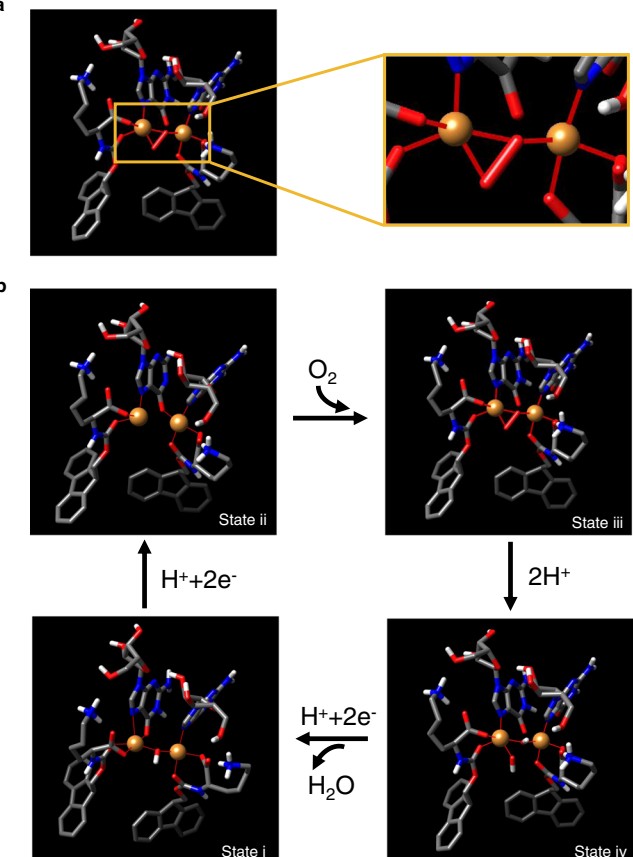

**Fig. 5 | Catalytic mechanism of the Fmoc-K/GMP/$Cu^{2+}$ complex. a** Density functional theory model of the dicopper/$O_2$ reactive intermediate. Inset illustrates the coordination bonding between the copper and O and N atoms. **b** Proposed catalyzed turnover by the dicopper-mimetic site, including the reduction of the dicopper site, reaction with $O_2$, cleavage of the peroxide bond and the reduction of $O_2$ to water. States i and iii were obtained from DFT calculations. N, O, C, H and Cu atoms are indicated in blue, red, dim gray, light gray and orange, respectively.

trinuclear copper sites (Supplementary Fig. 53), which resembles the peroxide intermediates of T2/T3Cu in natural multicopper oxidases[14]. We propose the role of GMP in the catalysis as follows: (i) The transformation of the copper site from Cu-O4 to Cu-O3(N), as a result of the assembly of GMP with Fmoc-K/$Cu^{2+}$, led to the enhancement of the activity. It has been widely reported that the reactivity of the metal center exhibits dependence on the coordinating atoms (e.g., C, O, N, P, Cl, S)[63–67]. (ii) The guanine base may assist the hydrogen transfer during the catalysis (State iii to iv), similar to $COO^-$[14,68], imidazole[69–72], etc. The lower turnover rate of the artificial complexes, in comparison to laccase, can be attributed to the following: (i) some of the copper ions that were added did not form the catalytically active T2-/T3-mimetic sites; (ii) the absence of T1Cu, which may significantly retard the decay rate of the intermediate conversion[14,73], and the redox potential of the copper cluster[16].

## Structural factors to the catalytic efficiency

Taking the experimental and simulation results above together, it is apparent that Fmoc-K self-assembly, and the heterogeneity of the molecular building blocks, were indispensable to the construction of the superior catechol oxidase-mimicking, $Cu^{2+}$-containing catalysts, which can be attributed to the following structural characteristics of complexes: First, the aromatic packing of the fluorenyl groups of Fmoc-K allowed the ordered distribution of the ligand groups

(carbonyl groups) around $Cu^{2+}$, which facilitated the copper clustering and prevented the coordination saturation of $Cu^{2+}$. This led to the higher activity of Fmoc-K/$Cu^{2+}$ ($k_{cat}$, 0.0232±0.006 $s^{-1}$) than that of GMP/$Cu^{2+}$ (~0 $s^{-1}$), Boc-K/$Cu^{2+}$ (~0 $s^{-1}$) and Cbz-K/$Cu^{2+}$ (~0 $s^{-1}$), and cooperation between Fmoc-modified amino acids and other ligands (e.g., nucleotides). Second, there was only an appropriate number of ligands from the rigid amino acid backbones of Fmoc-K to coordinate $Cu^{2+}$, leading to the potentially lower coordination number of $Cu^{2+}$ in the Fmoc-K aggregates, than that in Fmoc-R or Fmoc-H, that also can use the side chain group (guanidinium or imidazole) as the ligand. Therefore, the complexes containing Fmoc-K had much higher activity than those containing Fmoc-R or Fmoc-H. Third, the moieties of the heterogeneous ligand components exhibited comparable $Cu^{2+}$ binding capacity, which allowed the coordination of $Cu^{2+}$ to both components, leading to a synergy between the copper clustering effect (e.g., Fmoc-K) and proton transfer effect (e.g., nucleotides). Thus, GMP elicited a more remarkable enhancement of Fmoc-K/$Cu^{2+}$ activity, compared to other mononucleotides (e.g., CMP, AMP, UMP) which possess weaker $Cu^{2+}$ coordination capacity, or GDP and GTP that can extract $Cu^{2+}$ from the Fmoc-K. It is noteworthy that, in contrast to Fmoc-K, the activity of Fmoc-H/$Cu^{2+}$ can only be enhanced by GMP, and not CMP or AMP (Supplementary Fig. 54), which can be attributed to the strong Fmoc-H/$Cu^{2+}$ binding.

## Stability

The noncovalent bonds that maintain enzyme folding usually break down when heated, resulting in protein denaturation and activity loss. Native laccase showed optimal activity between 50 °C and 60 °C and fully deactivated at 80 °C (Supplementary Fig. 55). In contrast, the GMP/Fmoc-K/$Cu^{2+}$ complex showed thermophilic catalytic activity in the range of 20 °C to 90 °C, an almost 50-fold enhancement (Fig. 6a). At 80 °C, the oxidase activity of the complex was observed at as low as 20 nM $Cu^{2+}$ (Supplementary Fig. 56). The increase in temperature also led to enhancement of the activity of other $Cu^{2+}$ complexes (i.e., 126-fold for Fmoc-H/GMP/$Cu^{2+}$, 90-fold for Fmoc-R/GMP/$Cu^{2+}$ and 91-fold for Fmoc-K/guanosine/$Cu^{2+}$), with the exceptions of GMP/$Cu^{2+}$, Boc-K/GMP/$Cu^{2+}$, Fmoc-K/GTP/$Cu^{2+}$ and Cbz-K/GMP/$Cu^{2+}$ (Supplementary Fig. 57 and Fig. 6a). Fluorescence spectrometry of the complexes revealed that the catalytic assemblies were only partially dissociated or the aggregation states changed upon heating (Supplementary Fig. 58). It is likely that elevating the temperature resulted in less compact stacking of the fluorenyl rings and molecular association, along with enhanced mobility of the bond vibration. Thus, the requisite arrangement of the coordination atoms around $Cu^{2+}$ was facilitated. We also examined the Fmoc-K/GMP/$Cu^{2+}$ complex over different incubation times at 80 °C (Supplementary Fig. 59). About 60% of the activity of the artificial complex was retained after 5 h. Full deactivation of laccase was observed when the aggregate was incubated at its optimal temperature (60 °C) for 30 min. On the other hand, the catalytic complex exhibited reversible activity when alternating between 25 °C and 80 °C (and a 31.7% activity retention at both temperatures after 200 cycles; Supplementary Fig. 60), while laccase almost completely lost its activity after 18 cycles. Similarly, we investigated the response of the activities to acidification by cycling the pH between pH 6.0 and pH 2.6 (Supplementary Fig. 61). No loss of activity was found after over ten cycles, but laccase was completely inactive after 5 cycles. We also compared the storage stabilities of Fmoc-K/GMP/$Cu^{2+}$ and laccase maintained in aqueous solution or as lyophilized powder at room temperature. The solution-state copper complex showed a moderately higher retention of activity than laccase, while the powder-state complex did not show inactivation over as long as 60 days (Fig. 6b). We also investigated the tolerance of the Fmoc-K/GMP/$Cu^{2+}$ to high concentrations of salt (i.e., $NaNO_3$), and found the activity was even moderately increased when $NaNO_3$ reached 1 M (Supplementary Fig. 62).

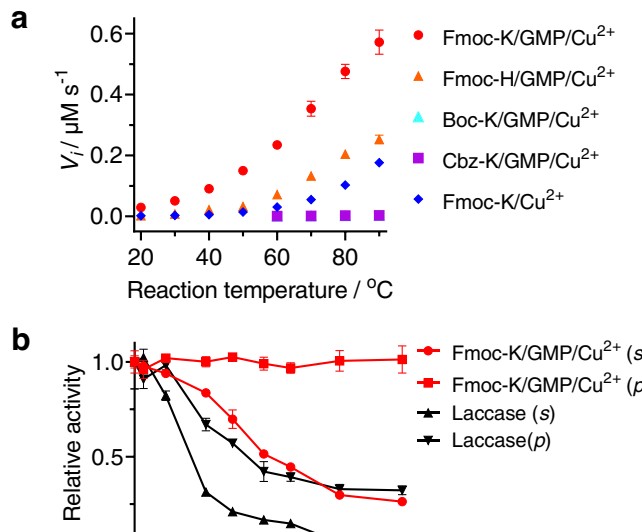

**Fig. 6 | Catalytic stability of the $Cu^{2+}$-containing complexes with 2,4-DCP as the reducing substrate. a** Temperature-dependent catalytic velocities of the various complexes. $[Cu^{2+}]$ = 0.3 μM. **b** Storage stability of the solution-state (*s*) and powder-state (*p*) of the copper complex and laccase at room temperature. $[Cu^{2+}]$ = 5 μM, [laccase] = 5 μM, [Fmoc-K] = 5 mM, [Fmoc-H] = 5 mM, [Boc-K] = 5 mM, [Cbz-K] = 5 mM, [GMP] = 10 mM, [laccase] = 5 μM, [2,4-DCP] = 1 mM, [4-AP] = 1 mM. The data in **a** and **b** are presented as the mean ± s.d., with the error bars representing the s.d. and *n* = 3 independent experiments.

## Discussion

In summary, here we report a simple strategy for the de novo design and fabrication of a supramolecular catalyst containing copper-cluster active sites and mimicking the catalytic functions of a catechol oxidase. The activities surpassed previously reported artificial catalysts and were significantly accelerated by increasing the reaction temperature. The catalysts also showed markable tolerance to temperature or pH cycling and room-temperature storage, compared to native enzymes. The superior thermophilic activities may enable the potential use of such enzyme mimics in the decolorization of textile wastewater, which is a common, high-temperature effluent system, and/or in the conversion of biomass (e.g., lignin) that can be softened and swelled at elevated temperatures. This work may pave the way for reconstructing versatile metallocluster active sites and expanding the application of the biocatalysts. These catalysts, based on the self-assembly of small molecules ($M_w$ < 410 Da), provide a putative model for primitive enzymes, whose thermophilic behaviors and environment-induced deactivation and activation may explain how these molecular assemblies survived under harsh, prebiotic conditions.

## Methods
### Materials

Fmoc-Lysine-OH·HCl(Fmoc-K), Fmoc-Histidine-OH(Fmoc-H), Fmoc-Arginine-OH(Fmoc-R), Boc-Lysine-OH(Boc-K), L-Lysine hydrochloride(K), Guanosine monophosphate(GMP), Adenosine monophosphate(AMP), Cytidine monophosphate(CMP), Uridine monophosphate(UMP), Guanosine, 2'-Deoxyguanosine 5'-monophosphate(dGMP), 4-Aminoantipyrine(4-AP), 3,3',5,5'-Tetramethylbenzidine(TMB), 2,6-Dimethoxyphenol(2,6-DMP), 3,5-Di-tert-butylcatechol(3,5-DTBC), Copper sulfate pentahydrate($CuSO_4·5H_2O$), Iron chloride hexahydrate($FeCl_3·6H_2O$), Cobalt chloride hexahydrate($CoCl_2·6H_2O$), Manganese chloride tetrahydrate($MnCl_2·4H_2O$), Magnesium chloride hexahydrate($MgCl_2·6H_2O$), Aluminum chloride($AlCl_3$), 2,4,6-Trichlorophenol

(2,4,6-TCP), Nickel chloride hexahydrate($NiCl_2 \cdot 6H_2O$), Zinc chloride($ZnCl_2$), 2,3,5,6-Tetrachlorophenol(2,3,5,6-TetraCP) and Boc-Histidine-OH(Boc-H) were purchased from Aladdin (China). Guanosine diphosphate(GDP), Guanosine triphosphate(GTP), Laccase and Tyrosinase were purchased from Yuanye (China). Cbz-Lysine-OH(Cbz-K), Pyrene, Ferrous chloride tetrahydrate($FeCl_2 \cdot 4H_2O$), Calcium chloride ($CaCl_2$), Cadmium Chloride($CdCl_2$), Chromic chloride hexahydrate ($CrCl_3 \cdot 6H_2O$), Scandium chloride hexahydrate($ScCl_3 \cdot 6H_2O$), 2,4-Dichlorophenol(2,4-DCP) and Vanadium chloride($VCl_4$) were purchased from Innochem (China). Water was deionized using a Milli-Q system($\geq 18.25$ MΩ·cm$^{-1}$). Fmoc-K, Fmoc-H, Fmoc-R, Boc-K, Cbz-K, Boc-H, Cbz-H, K and H (0.1 M) were dissolved in ultrapure water in the presence of 0.1 M hydrochloric acid. The solutions were kept for over 2 weeks priror to further use.

## Instruments
UV-Vis absorption spectra were recorded using a UV-2600 spectrometer equipped with a temperature-control accessory (Shimadzu). Fluorescent emission spectra were recorded using a G9800A fluorescence spectrophotometer with a temperature-control accessory (Agilent Technologies). SEM characterization was conducted using a Supra microscope at 5.0kv (Zeiss). TEM characterization was conducted on a Hitachi 7800 microscope in bright-field mode at 80kV. SAED were carried out on a JEM-F200 system with cooling stage and energy-dispersive X-ray spectrometer (JEOL). AFM characterization was conducted on a Nano Wizard 4 BioScience (JPK). CD spectra were recorded with a J-815 spectropolarimeter (Jasco) under the following conditions: optical path, 0.5mm, bandwidth, 10nm, scan speed, 50nm/min. EPR measurements were conducted on an ELEXSYS-II E500 EPR spectrometer with a low-temperature accessory (Bruker BioSpin). Three-pulse EPR measurements were conducted on an EPR100 X-band pulsed EPR spectrometer (CIQTEK) with a variable temperature controlled dry liquid helium-free cryogenic system (CIQTEK). Powder XRD patterns were collected on a Smartlab-9KW diffractometer equipped with a copper filter (Rigaku) under the following conditions: scan speed, 5 min$^{-1}$, Cu Kα radiation, λ = 1.54 Å, temperature, 120 K.

XAFS were collected at 1W1B station in Beijing Synchrotron Radiation Facility (BSRF) with fluorescence mode. The acquired EXAFS data were processed using the ATHENA module implemented in the IFEFFIT software packages[74]. 1H NMR were recorded in DMSO-d6 on Bruker AVANCE III HD 400 MHz. The true concentrations of enzyme were conducted using an XII ICP-MS (ThermoFisher) after microwave digesting on a Mars 6 equipment (CEM).

## Sample preparation for characterization
(1) For obtaining SEM images, we applied the self-assembled sample solution onto a silicon wafer and allowed it to settle for 20 min. Any unbound sample was then removed by wicking it away. Prior to observation, a thin layer of platinum particles was applied to the surface to ensure sample conductivity by the gold spray instrument with 15 s. (2) To prepare the sample for TEM imaging, we deposited the self-assembled solution onto a hydroxylated copper sheet and allowed it to sit for 20 min. Excess solution was removed using filter paper, and the remaining solution was left to evaporate. (3) For AFM imaging, 5 μL of the solution was carefully placed onto freshly cleaved mica surfaces. This was followed by a gentle wash with double distilled water and subsequent drying under a nitrogen flow.

## Activity assay
The amino acid amphiphiles (Fmoc-K, Fmoc-H, Fmoc-R) were dissolved in ultrapure water to prepare 100 mM stock solutions, which were stored for over 2 weeks before use. Subsequently, the required concentrations of amino acid amphiphiles, nucleotide, and copper ions were added to ultrapure water. The mixture was then incubated at room temperature for 12 h. Using 2,4-DCP/4-AP or other substrates

(with a molar extinction coefficient of 13.6 mM$^{-1}$ cm$^{-1}$ for the adduct of the oxidized 2,4-DCP with 4-AP)[75,76], the reactions were conducted at ~pH 6.00. Time-dependent absorbance changes were recorded, and this data was used to determine the initial catalytic velocity ($V_i$) and apparent kinetic parameters.

## Fluorescence assay
The amino acid amphiphiles (Fmoc-K, Fmoc-H, Fmoc-R) were dissolved in ultrapure water to prepare 100 mM stock solutions, which were stored for over 2 weeks. Subsequently, the required concentrations of amino acid amphiphiles, nucleotide, and copper were added to ultrapure water. The mixture was then incubated at room temperature for 12 h. The experimental conditions involved an excitation wavelength of 290 nm and an emission wavelength ranging from 300 nm to 350 nm, as well as an excitation wavelength of 320 nm and an emission wavelength ranging from 350 nm to 450 nm. The scan rate was set at 120 nm/min.

## Pyrene-involved experiment
To prepare the pyrene stock solution, 0.5 mg of pyrene was dissolved in 5 mL of methanol. For the fluorescence assay, the sample was mixed with the pyrene stock solution in a volume ratio of 100:1 immediately before analysis. The experimental conditions were set as follows: an excitation wavelength ($\lambda_{ex}$) of 334 nm, an emission wavelength ($\lambda_{em}$) ranging from 360 nm to 400 nm, and a scan rate of 120 nm/min. Two distinct fluorescent peaks, labeled as $I_1$ and $I_3$, were observed at 373 nm and 384 nm, respectively. The intensity ratio ($I_1/I_3$) between these two peaks can be utilized to investigate the polarity of the microenvironments within the self-assembled complexes.

## Deaeration treatment
The catalyst solution and substrate solution were placed in separate containers equipped with rubber stoppers. The Schlenk system was employed to evacuate air from the containers and replace it with argon gas, a procedure that was repeated three times. Subsequently, a continuous flow of argon was maintained for 30 min. For activity assays or UV-vis spectra measurements, a needle was utilized to rapidly transfer the substrates into the sample solution.

## Continuous-Wave Electron Paramagnetic Resonance (cw-EPR) spectroscopy
The samples were transferred to EPR tubes and frozen using a liquid nitrogen atmosphere for subsequent EPR experiments. cw-EPR spectra were collected using a Bruker ELEXSYS-II E500 CW-EPR spectrometer equipped with an Air Products cryostat and temperature controller modified for nitrogen gas flow sample cooling. The spectra were obtained under the following conditions: microwave frequency set at 9.41 GHz, microwave power at 10 mW, modulation amplitude at 2 Gauss, modulation frequency at 100 kHz, sweep time at 120 s, and temperature at 77 K. To ensure accuracy, a minimum of 5 scans were recorded and averaged for each spectrum. The simulations were conducted using the MATLAB R2021a toolbox Easyspin 5.2.35 by the function 'Pepper'.

## Electron Spin Echo Envelope Modulation (ESEEM) spectroscopy
The samples were transferred to 4 mm outer diameter quartz EPR tubes and frozen using a liquid helium atmosphere (6K) for subsequent ESEEM experiments. Three-pulse ESEEM spectra were collected using a CIQTEK EPR100 spectrometer equipped with a liquid helium circulation system for helium gas flow sample cooling. The signal was recorded by measuring the stimulated electron spin echo intensity as a function of T using the pulse sequence: (π/2) - τ - (π/2) - T - (π/2) - τ - echo on the Cu-complex, where τ is the dephasing time and T is the waiting time. The duration of the (π/2) pulses for the Cu-complex is indicated in the figure captions. The interpulse delay τ was fixed at

200 ns. The time interval T was incremented from 100 ns with a step size of 16 ns, resulting in a total of 256 points. The microwave power was set to 200 mW, and at least 100 waveforms were accumulated for each sample. The envelope modulation was cosine Fourier transformed to generate ESEEM frequency spectra. Data processing and analysis were performed using EPR Data Processing (Version 4.1.6, CIQTEK), and simulations were conducted using the MATLAB R2021a toolbox Easyspin 5.2.35 by the function 'saffron'.

### Estimation of hyperfine constant from NMR shift

When dealing with a paramagnetic complex, the NMR chemical shifts of the ligand nuclei experience a change $\Delta_{CS} = \Delta_{HFI} + \Delta_{Dia} + \Delta_{PC}$. The three terms correspond to the hyperfine, diamagnetic and pesudo-contact shifts. The first term, due to the hyperfine (magnetic) field generated by the unpaired electron(s), is typically 2–3 orders of magnitude larger than the latter two when the paramagnetic center is close to the nucleus concerned. Therefore, as a crude approximation $\Delta_{CS} \approx \Delta_{HFI}$. It's worth noting that NMR relaxation is typically much slower than EPR, which means that the field that the nuclei "feel" from the unpaired electron(s) is an average value over time. From the hyperfine interaction Hamiltonian

$$\hat{H}_{HFI} = \hat{S} \cdot \boldsymbol{A} \cdot \hat{I}$$

Averaging the electron spin (denoting the average by angle brackets), it becomes

$$\hat{H}_{HFI} = \left\langle \hat{S} \right\rangle \cdot \boldsymbol{A} \cdot \hat{I}$$

Because the molecules in a solution are tumbling quickly, the apparent chemical shift corresponds to an angle-averaged value. In a strong external magnetic field in the z direction such as that in an NMR instrument, the spin averages to zero in the x and y directions, and thus the above simplifies to $\left\langle S_z \right\rangle A_{iso} I_z$. Therefore, the absolute value of the hyperfine shift for an $I = \frac{1}{2}$ ($I_z = \pm \frac{1}{2}$) nucleus, such as $^1H$, is

$$\Delta_{CS} = \hat{H}_{HFI}\left(I_z = \frac{1}{2}\right) - \hat{H}_{HFI}\left(I_z = -\frac{1}{2}\right) = \left\langle S_z \right\rangle A_{iso} \times \left[\frac{1}{2} - \left(-\frac{1}{2}\right)\right] = \left\langle S_z \right\rangle A_{iso}$$

or

$$A_{iso} = \Delta_{CS} / \left\langle S_z \right\rangle$$

For a 400 MHz NMR, the magnetic field is 9.4 T. The EPR transition frequency for a $g = 2.1$ species in such a magnet is $2.1 \times 9.4 \times 10^4 \text{gauss} \div 714.48 \frac{\text{gauss}}{\text{GHz}} = 276$ GHz. (714.48 gauss/GHz is Planck's constant divided by Bohr magneton). Per Boltzmann distribution, the ratio of populations of $S_z = 1/2$ to $S_z = -1/2$ electrons, at 298 K, is 0.9566. Therefore, the weighted average of $S_z$ is

$$\left\langle S_z \right\rangle = \frac{\frac{1}{2} \times 0.9566 + \left(-\frac{1}{2}\right) \times 1}{0.9566 + 1} = -0.0111$$

Under the conditions where the NMR spectra were taken, and assuming labile $Cu^{2+}$ binding, each guanosine site has a $Cu^{2+}$ occupancy of only 0.01. Therefore, the average hyperfine field felt by the guanosine ligand should be scaled down accordingly, to $\left\langle S_z \right\rangle = 1.11 \times 10^{-4}$. As shown in Supplementary Fig. 19c, the NMR peak of the proton moved by about 0.05 ppm, or $\Delta_{CS} = 20$ Hz in a 400 MHz magnet upon adding $Cu^{2+}$. Therefore, one can estimate that

$$A_{iso} = \frac{\Delta_{CS}}{\left\langle S_z \right\rangle} = 20\text{Hz} \div 1.11 \times 10^{-4} = 0.18\text{MHz}$$

### Simulation

In order to gain a better understanding of the active site, a comprehensive set of quantum chemical (QM) calculations were performed. These calculations aimed to investigate the structures and interactions of peptide-assembled nanostructures, GMP, $Cu^{2+}$, and O2. Specifically, the crystal structures of Fmoc-K and Fmoc-H, as well as the co-assembly of GMP and $Cu^{2+}$, were analyzed using first-principles calculations. The Vienna ab initio simulation package 5.4.4 and the plane-wave pseudopotential method were employed for all the calculations[77,78]. The Perdew–Burke–Ernzerhof functional[79] with the generalized gradient approximation was used to account for electronic exchange and correlation effects. The full-potential projector augmented wave method was employed to describe the behavior of the core electrons[80,81]. A plane-wave expansion with an energy cut-off of 400 eV was implemented, ensuring that the force on the relaxed atoms remained below 0.03 eVÅ$^{-1}$. The Monkhorst–Pack scheme was utilized to generate a k-point grid for sampling the Brillouin zone, with a grid size of $(8 \times 5 \times 1)$ specifically applied to Fmoc-K and Fmoc-H. The peptide conformations obtained from the first-principles calculations were employed in the QM calculations. Additionally, Gaussian 09 software was utilized to optimize the Cu-dimer and triple clusters at the B3LYP/6-31G(d,p) level of theory[82–84]. CrystalDiffract offers robust simulation capabilities for both x-ray and neutron powder diffraction. In this study, the constant-wavelength X-ray simulation type was utilized to generate the XRD patterns. This simulation type was chosen from the available options, which include constant-wavelength X-r.ays, neutrons, energy-dispersive X-rays, and time-of-flight neutrons.

### Statistical analysis

The statistical analysis methods for quantitative data are described in the results sections or figure captions. All values are presented as mean ± standard deviation (s.d.) with the sample size indicated. All statistical tests were two-sided, and the GraphPad Prism 8 software was utilized for the statistical analyses.

### Reporting summary

Further information on research design is available in the Nature Portfolio Reporting Summary linked to this article.

## Data availability

All data are available within the main text or the Supplementary information. PDB: 1GYC, PDB: 1AOZ.

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

## Acknowledgements

The authors are grateful for financial support from the National Science Foundation China (21872044, 52173194), Beijing Natural Science Foundation (2232017), and Fundamental Research Funds for the Central Universities (XK1806, buctrc201902). The theoretical simulations are supported by Hefei advanced computing center and high performance computing platform of BUCT. We also thank Prof. L. Jiang from Institute of Chemistry, CAS, for assistance in SAED measurement, Prof. Y. Liu from the Technical Institute of Physics and Chemistry, CAS, for the EPR measurement, and Prof. L. Zheng from Institute of High Energy Physics, CAS for the XANES measurement.

## Author contributions

Z.-G.W. conceived and designed the experiments. S.C.X., S.Y.L. and Z.-G.W. performed the experiments. P.D.D. provided suggestions on the experimental details. S.C.X., Z.-G.W. and H.W. collected and analysed the data. H.W. and X.H.S. performed the theoretical simulations. H.J.Y. provided supports on ESEEM operations and analysis. J.K.L. provided supports on EPR/ESEEM analysis and NMR analysis. H.F.W. provided supports on the EPR/ESEEM analysis. W.J.X, S.M.C., and L.S. provided supports on the analysis of XAS/EXAFS data. Z-.G.W. supervised the project. Z.-G.W., H.W. and S.C.X. wrote the manuscript. All authors discussed the results and commented on the manuscript.

## Competing interests

The authors declare no competing interests.
