## [Peer Review File · Nature Communications]

REVIEWER COMMENTS

Reviewer #1 (Remarks to the Author):

The manuscript by Wang et al. describes a supramolecular mimetic metalloenzyme with good catalytic efficiency, even approaching that of the native enzyme. The authors provided substantial experimental and theoretical results through EPR, DFT et al. for a comprehensive study of catalytic site structures and structure-activity relationship. All previous comments were well responded to, despite of some minor issues to be addressed. The structure from the crystal provided useful molecular insight. Based on the current version, I recommend acceptance of this manuscript to Nature Communications.

1. The authors used several different concentration ratios of the components in different experiments (e.g., in SEM, EPR and EXAFS). Would the different ratios affect the coordination and the structures of the complexes?

2. The authors stated that at 100 μM Cu^{2+} , the V_i value was $5.264 \pm 0.253 \mu\text{M s}^{-1}$ for the Fmoc-K/GMP/ Cu^{2+} , while as shown in Figure 4b, the value was lower than 5. Please explain this.

3. There are two spectra for CMP/ Cu^{2+} in Figure S12. Please correct it.

4. The introduction is not very clear. Why did the authors choose supramolecular self-assembly? What's unique about noncovalent interactions? Why did the authors choose Fmoc rather than other aromatic motifs to initiate self-assembly? Answering these questions is important for readers to better understand the background information and design rationale.

Reviewer #2 (Remarks to the Author):

The manuscript is improved relative to the original submission and it could be in principle suitable for Nat Commun. The improvements have been made in complex characterization, but questions remain regarding kinetic studies. Specifically, SI doesn't contain enough information (extinction coefficients of the products, etc) for me to go over the results and check if there is sufficient catalyst turnover, etc. This needs to be clarified before acceptance.

Reviewer #3 (Remarks to the Author):

In the following I would like to comment on the employment of EPR spectroscopy in the manuscript NCOMMS-23-03097-T "A Supramolecular Metalloenzyme Possessing Robust Oxidase-Mimetic Catalytic Function", by Xu et al. The present manuscript represents a revised version of work originally submitted to [other nature journal] and reviewed by me for the use of EPR spectroscopy.

I originally suggested that the authors should perform more accurate data analysis of the obtained EPR spectra using modern simulation software and, in addition, appropriate hyperfine spectroscopy to study the nitrogen coordination of the complexes they are studying.

I acknowledge that the quality of the data interpretation was improved by using the EasySpin simulation software. Thereby, the authors managed to derive g values and Cu hyperfine values for the studied complexes. Yet, a more in-depth of the spectral line shape, with respect to resolved ^{14}N hyperfine splittings in particular in the well resolved $g//$ region is still missing. In addition, it was not clear to me if the authors simulated the spectra with Cu in natural abundance or only for the ^{63}Cu isotope. The authors in addition, performed ESEEM spectroscopy to study the hyperfine interaction of the electron spin on the copper with surrounding nitrogen atoms. Unfortunately, also from ESEEM spectroscopy no clear information on the coordinating nitrogens could be obtained. This limits the information content which can be drawn from EPR and ESEEM spectra with respect to the structure and coordination environment, beyond the qualitative picture of the coordination geometry provided by the authors.

While inspecting the EPR spectra it appeared to me that the spectra exhibit saturation artifacts, which can distort the line shape and blur spectral features. These artifacts most probably originate from the very high microwave powers applied in the EPR experiments. The applied 10 mW of power almost certainly saturates the spectra. Lower powers may lead to better resolved spectra, better match with simulations and finally provide information on the coordinating nitrogens.

Reviewer #4 (Remarks to the Author):

The manuscript has been revised significantly according to all reviewers' comments, therefore it could be published in Nat. Commun. as it is.

Reviewer 1

The manuscript by Wang et al. describes a supramolecular mimetic metalloenzyme with good catalytic efficiency, even approaching that of the native enzyme. The authors provided substantial experimental and theoretical results through EPR, DFT et al. for a comprehensive study of catalytic site structures and structure-activity relationship. All previous comments were well responded to, despite of some minor issues to be addressed. The structure from the crystal provided useful molecular insight. Based on the current version, I recommend acceptance of this manuscript to Nature Communications.

Response: We are grateful for the reviewer's approval. We have revised the manuscript following the detailed suggestions. Please check the specific responses.

Specific comments:

1. The authors used several different concentration ratios of the components in different experiments (e.g., in SEM, EPR and EXAFS). Would the different ratios affect the coordination and the structures of the complexes?

Our Response:

We would like to express our gratitude to the reviewer for their insightful comment. We agree that the reviewer's concern is entirely reasonable. The different characterization techniques involved in our study have distinct criteria regarding the molar ratios and concentrations of the components, particularly the ligands to Cu²⁺. Regarding the activity assay, we found that if the concentration of Cu²⁺ is too high (e.g., >100 μM), the absorbance of the product quickly exceeds the detection limit. Therefore, we used 5 μM of Cu²⁺ for measuring the activity, which should be suitable. Additionally, we observed that the activity was approximately 25 times higher at 90°C than at room temperature. As a result, we chose 0.3 μM Cu²⁺ to investigate the effect of temperature. In contrast, cw-EPR measurements required a higher concentration of Cu²⁺ (e.g., 100 μM) to obtain spectra with a high signal-to-noise ratio, whereas the

ESEEM measurements required an even higher concentration of Cu²⁺. Finally, for XANES/EXAFS measurements, we determined that the copper should reach at least 0.1% of the total mass.

To meet various characterization requirements and ensure reliable outcomes, we utilized multiple concentration ratios of the components in different experiments. We are confident that the varying ratios of Cu²⁺ to ligands employed in these measurements did not impact the coordination or structure of the catalytic complexes. Our findings indicate that within a specific range of Cu²⁺ concentrations (0.5~100 μM, as shown in Figure 4), the V_i values of the complexes were directly proportional to the concentration of Cu²⁺. Furthermore, the concentrations of ligands used in our experiments were significantly higher (~5-10 mM) than that of Cu²⁺ (< 200 μM). These observations suggest that there were ample coordination sites available to bind with Cu²⁺ and form oxidase-mimetic active sites, which possess similar structures of coordination spheres.

To summarize, we believe that the various ratios will not have an impact on the coordination or structure of the complexes.

2. The authors stated that at 100 μM Cu²⁺, the V_i value was $5.264 \pm 0.253 \mu\text{M s}^{-1}$ for the Fmoc-K/GMP/Cu²⁺, while as shown in Figure 4b, the value was lower than 5. Please explain this.

Our Response:

We are grateful for pointing out this mistake. The V_i value is corrected into $4.675 \pm 0.363 \mu\text{M s}^{-1}$ on the Page 23, Line 2 of the revised manuscript.

3. There are two spectra for CMP/Cu²⁺ in Figure S12. Please correct it.

Our Response:

We thank the reviewers for pointing out our mistakes. The spectrum of Fig. S12(c) should be AMP/Cu²⁺. We have revised this in the supplementary information.

4. The introduction is not very clear. Why did the authors choose supramolecular self-assembly? What's unique about noncovalent interactions? Why did the authors choose Fmoc rather than other aromatic motifs to initiate self-assembly? Answering these questions is important for readers to better understand the background information and design rationale.

Our Response:

We are grateful for the insightful suggestion. We have revised introduction in the manuscript.

Page 4 Line 15 of the revised manuscript, “Native enzymes rely on well-defined tertiary structures, which are formed through noncovalent interactions in the protein chain, to organize essential functional groups in a pocket where the active sites are created. We were inspired to arrange ligands in an ordered manner, via designed molecular self-assembly (or folding), for clustering metals. The noncovalent interactions among the molecular building block may allow the self-assemblies to have a great structural flexibility, similar to native enzymes, which facilitates the access of the molecular substrates to the active sites inside the supramolecular entity.”

Page 5 Line 8 of the revised manuscript, “The fluorenyl moiety with an ortho-fused tricyclic structure may stack in a directional manner, which allows for the ordered arrangement of ligand groups from the side chain or backbones of the amino acids, for creating coordinatively unsaturated copper centers.”

Reviewer 2

The manuscript is improved relative to the original submission and it could be in principle suitable for Nat Commun. The improvements have been made in complex characterization, but questions remain regarding kinetic studies. Specifically, SI doesn't contain enough information (extinction coefficients of the products, etc) for me to go over the results and check if there is sufficient catalyst turnover, etc. This needs to be clarified before acceptance.

Our Response:

We thank reviewer for the suggestion. The references (2,3) of the revised supplementary materials (*Spectrochim Acta A* **2022**, 281, 121606; *Catal. Lett.* **2017**, 147, 2144-2152.) have shown that the molar extinction coefficients of the adduct of the oxidized 2,4-DCP (mostly commonly used substrate in this work) with 4-AP is $13.6 \text{ mM}^{-1} \text{ cm}^{-1}$. Herein, there is an issue with estimating the turnover number. Based on the time-dependent absorbance changes shown in Figure S34, we observed that in the presence of Fmoc-K/GMP/Cu²⁺ (with Cu²⁺ at a concentration of 5 μM) and 2,4-DCP (at concentrations of 1 mM)/4-AP (at concentrations of 1 mM), the absorbance level reached the highest value detectable by our UV-vis spectrophotometer (Shimadazu, UV-2600) (~ 4) after the catalyzed reaction had proceeded for approximately 10 minutes. At this point, around 307 μM of the red adduct was generated. However, it is apparent that there could still be more products being formed, but their quantity cannot be monitored. This implies that even if 100% of the 2,4-DCP (1 mM) was converted, the products' absorbance may exceed the detection range of the UV-vis spectrophotometer. On the other hand, we observed that a single copper ion can catalyze the oxidation of at least around sixty 2,4-DCP molecules, which demonstrates that Fmoc-K/GMP/Cu²⁺ is indeed a catalyst.

Reviewer 3

In the following I would like to comment on the employment of EPR spectroscopy in the manuscript NCOMMS-23-03097-T “A Supramolecular Metalloenzyme Possessing Robust Oxidase-Mimetic Catalytic Function”, by Xu et al.

The present manuscript represents a revised version of work originally submitted to [redacted] and reviewed by me for the use of EPR spectroscopy. I originally suggested that the authors should perform more accurate data analysis of the obtained EPR spectra using modern simulation software and, in addition, appropriate hyperfine spectroscopy to study the nitrogen coordination of the complexes they are studying.

I acknowledge that the quality of the data interpretation was improved by using the Easypin simulation software. Thereby, the authors managed to derive g values and Cu hyperfine values for the studied complexes. Yet, a more in-depth of the spectral line shape, with respect to resolved ^{14}N hyperfine splittings in particular in the well resolved g// region is still missing. In addition, it was not clear to me if the authors simulated the spectra with Cu in natural abundance or only for the ^{63}Cu isotope. The authors in addition, performed ESEEM spectroscopy to study the hyperfine interaction of the electron spin on the copper with surrounding nitrogen atoms. Unfortunately, also from ESEEM spectroscopy no clear information on the coordinating nitrogens could be obtained. This limits the information content which can be drawn from EPR and ESEEM spectra with respect to the structure and coordination environment, beyond the qualitative picture of the coordination geometry provided by the authors.

While inspecting the EPR spectra it appeared to me that the spectra exhibit saturation artifacts, which can distort the line shape and blur spectral features. These artifacts most probably originate from the very high mw powers applied in the EPR experiments. The applied 10 mW of power almost certainly saturates the spectra. Lower powers may lead to better resolved spectra, better match with simulations and finally provide information on the coordinating nitrogens.

Our Response:

We are grateful for the valuable comments. We simulated the spectra with Cu in natural abundance by using Easypin. High powers of EPR may indeed distort the line shape and blur spectral features, as the reviewer concerned. Accordingly, we performed EPR experiments at different powers (from 1 mW to 10 mW) for Fmoc-K/GMP/Cu²⁺. As shown in Figure R1, the spectra at different powers exhibited similar shape, but difference in intensity. The spectrum under 10 mW is about 3.16-fold ($\sqrt{10}$) of that under 1 mW, consistent with the expected $I_{EPR} \propto \sqrt{P}$ scaling without power saturation. After rescaling the EPR spectra collected at different powers to achieve identical intensities (Figure R1, inset), their shapes almost completely overlapped. Moreover, the

hyperfine splitting was clearly observed for CMP/Cu²⁺ (Figure S12a) under this power. These facts indicate that 10 mW power applied for the EPR experiment was reasonable. We would like to mention that our CW-EPR experiments were conducted at 77 K, and it is worth noting that while a typical Cu²⁺ species at liquid helium temperatures can be easily saturated with 10 mW, a higher saturating power is expected in our experimental conditions

Figure R1. EPR spectra of Fmoc-K/GMP/Cu²⁺ at different powers. Inset: The EPR spectra that were collected under different powers were rescaled to achieve identical intensities.

To resolve the ¹⁴N hyperfine splittings, we attempted to take the derivative of the spectral lines to obtain the second-derivative spectra. However, we still did not observe the hyperfine splittings in all the complexes but CMP/Cu²⁺ (which we did observe without any derivation treatment) (Figure S12a). This may result from the structural distributions in the coordination spheres, i.e. the EPR lines broaden due to g-strains and A-strains. (*Multifrequency Electron Paramagnetic Resonance*, edited by Sushil K. Misra, chapter 2, Page 35) As far as we know, it is not uncommon for Ni and Cu complexes to exhibit field-dependent obscuring of hyperfine lines at X- or higher microwave bands.

We acknowledge that ESEEM may not be a powerful technique to gather

information regarding the direct coordination of copper to ^{14}N , particularly N7 of guanine, which belongs to the first coordination sphere. However, the ESEEM findings have eliminated the likelihood of the side chain amine of Fmoc-K being engaged in copper coordination. For examining the direct coordination of metals (Cu^{2+}) to magnetic nuclei, ENDOR is a more appropriate method, but it necessitates a metal ion concentration of up to mM level (W. Hagen, *Biomolecular EPR Spectroscopy*, © 2009 by Taylor & Francis Group, LLC, chapter 14, page 227). We have noticed that such a concentration of Cu^{2+} led to the precipitation of the Fmoc-K/GMP/ Cu^{2+} complexes. Utilizing continuous wave or pulse wave EPR alone may not be sufficient to gain a comprehensive understanding of the coordination environment of copper.

To gain further insights into the coordination environment of Cu^{2+} in Guanosine/ Cu^{2+} , we conducted ^1H -NMR measurements. The interaction between Cu^{2+} and the ligand can cause a faster relaxation of nearby ^1H nuclei, resulting in resonance line broadening. In Figure R2 (a) (Figure S19 (a) of the revised manuscript), it can be observed that the addition of Cu^{2+} resulted in broadening of the line at approximately 7.93 ppm, which corresponds to the hydrogen on C8 of the guanine ring. This finding confirms that N7 of guanine is directly coordinated to Cu^{2+} . It is worth noting that there was no line broadening observed for the hydrogens of Fmoc-K (Figure R2 (b), Figure S19 (b) of the revised manuscript), indicating that these hydrogens were not in close proximity to the copper center. This suggests that the copper was instead coordinated to the carboxylate and carbonyl groups. Additionally, similar line broadening of the hydrogen on C8 of the guanine ring was also observed for Fmoc-K/Guanosine after the addition of Cu^{2+} . (Figure R2 (c), Figure S19c of the revised manuscript).

The NMR peaks of the ligands in a paramagnetic complex are shifted by the hyperfine (magnetic) field from the unpaired electron(s). This phenomenon can be clearly observed in our experiment. When the nucleus concerned is located near the paramagnetic center, the hyperfine constant tends to be in the range of 103 to 105 ppm, on the order of MHz. This value takes precedence over the pseudocontact shift and diamagnetic shift, both of which are on the order of 1 ppm or kHz. Even when accounting for the thermal averaging of S_z , which is around 0.01 at room temperature,

the hyperfine shift remains dominant. (See Ref. (49) of the revised manuscript.) So, we can estimate the ^1H hyperfine constant from solution NMR shift. NMR relaxation is typically slower than EPR or molecular tumbling. For a 400 MHz NMR used in this manuscript, the magnetic field is 9.4 T. The EPR transition frequency for a $g = 2.1$ species in such a magnet is $2.1 \times 9.4 \times 10^4 \div 714.48$ (Planck constant/Bohr magneton) = 276 GHz. It is estimated with Boltzmann distribution that the ratio of populations of $S_z = 1/2$ to $S_z = -1/2$ electrons, at 298 K, is 0.9566. Therefore, the weighted average of S_z is -0.0111. Assuming that the binding of Cu^{2+} to the guanosine sites is labile and involves frequent binding and unbinding events during the NMR relaxation time of approximately 50 milliseconds, a molar ratio of 1:100 of Cu^{2+} to guanosine would result in a Cu^{2+} occupancy of 0.01 for each guanosine site. As a result, the *average* hyperfine field on each ligand is scaled by a factor of 0.01, reducing $\langle S_z \rangle$ to 1.11×10^{-4} . The NMR peak of the C8 proton of guanine ring changed by about 0.05 ppm, or $\Delta_{\text{CS}} \approx \Delta_{\text{HFI}} = 20$ Hz in a 400 MHz magnet upon adding Cu^{2+} . $A_{\text{iso}} = \Delta_{\text{CS}} / \langle S_z \rangle = 20 \text{ Hz} \div (1.11 \times 10^{-4}) = 0.18$ MHz.

According to the reference (49) of the revised manuscript (*J. Am. Chem. Soc.* **2000**, 122, 3701-3707.), Cu^{2+} -bound histidines in blue copper proteins have an A_{iso} value of ~ 1.5 MHz for H_δ . The histidines above typically bind the copper in the $d_{x^2-y^2}$ direction, where the unpaired electron is located, in a $3d^9$ Cu^{2+} complex. According to Schweiger et al (48) (this is reference number of the revised manuscript, *J. Phys. Chem. A* **1999**, 103, 5446-5455.), when the aromatic N ligand binds from a direction away from the unpaired electron, e.g., d_z^2 in Cu^{2+} complexes and $d_{x^2-y^2}$ in Co^{2+} complexes, the unpaired electron population on the ligand (hence the A_{iso} value) decreases by approximately one order of magnitude. We believe that in Guanosine (or GMP)/ Cu^{2+} , the ligand binds in the d_z^2 direction, away from the unpaired electron. Therefore, an A_{iso} of 0.18 MHz is reasonable. It also agrees with our ESEEM results that no significantly split ^1H peaks show up. Also, the reference (49) (*J. Am. Chem. Soc.* **2000**, 122, 3701-3707.) of the revised manuscript shows that, by moving the ^1H one bond away from the Cu^{2+} (Cys H_β vs His H_δ), the A_{iso} reduces by approximately an order of magnitude as well.

Taken together the results of $^1\text{H-NMR}$, ESEEM and cwEPR, it can be concluded that the first coordination sphere of Cu^{2+} in Fmoc-K/GMP/ Cu^{2+} was composed of N7 of guanine base and carbonyl/carboxylate of the lysine moiety, which provides experimental evidence for the theoretical simulation.

Figure R2. $^1\text{H-NMR}$ of (a) Guanosine, Guanosine/ Cu^{2+} . [Guanosine] = 10 mM, [Cu^{2+}] = 50 μM . (b) Fmoc-K, Fmoc-K/ Cu^{2+} . [Fmoc-K] = 10 mM, [Cu^{2+}] = 50 μM . (c) Fmoc-K/Guanosine, Fmoc-K/Guanosine/ Cu^{2+} . [Fmoc-K] = 5 mM, [Guanosine] = 5 mM, [Cu^{2+}] = 50 μM .

The discussion on the $^1\text{H-NMR}$ results has been added to Page 16 Line 19 of the revised manuscript, “To gain further insight into the coordination of Cu^{2+} to the guanine ring, we conducted hydrogen nuclear magnetic resonance ($^1\text{H-NMR}$) measurements of guanosine/ Cu^{2+} in DMSO-d_6 . Line broadening and shifting are mainly observed on the NMR signals of hydrogen nuclei that are in close proximity to the paramagnetic Cu^{2+} ions. This is due to the interaction between the magnetic field of the unpaired electron of Cu^{2+} and the nearby NMR nucleus, which causes the relaxation time of those nuclei

to become shorter.⁴⁷ As shown in Figure S19, after adding Cu^{2+} , a broadening and shifting of the peak at ca. 7.93 ppm was observed, which was attributed to the hydrogen on C8 of guanine ring. This finding confirms the coordination of N7 of guanine to Cu^{2+} . The hyperfine constant, A_{iso} , is estimated to be 0.18 MHz from ^1H -NMR shift (Details of the analysis and calculation can be found in supplementary information). No line broadening was observed for the hydrogens of Fmoc-K/ Cu^{2+} (Figure S19b), indicating the hydrogens were away from Cu^{2+} . A similar line broadening of the guanine ring was also observed for Fmoc-K/guanosine after adding Cu^{2+} (Figure S19a). Based on the results of ^1H -NMR, cwEPR, and ESEEM, it can be reasoned that the coordination sphere of Cu^{2+} in Fmoc-K/guanosine/ Cu^{2+} (or Fmoc-K/GMP/ Cu^{2+}) was composed of N7 of guanine base and carbonyl/carboxylate of the lysine moiety.

”

Reviewer 4

The manuscript has been revised significantly according to all reviewers' comments, therefore it could be published in Nat. Commun. as it is.

Our Response:

We are very grateful to the reviewer for approving our work.

REVIEWERS' COMMENTS

Reviewer #1 (Remarks to the Author):

The authors have addressed my concerns, and I support the acceptance of this work.

Reviewer #2 (Remarks to the Author):

With some turnover established, it's good to see that the reported assemblies are in fact catalytic and the reported kinetic parameters are valid. I'm puzzled by the authors' response that high absorbance in their kinetic assay precludes the proper turnover determination, though. What happened to dilution and/or shorter pathway cuvettes? It's not really essential, as TN of 4 is ok

Reviewer #3 (Remarks to the Author):

In the latest revised version of the manuscript "A Supramolecular Metalloenzyme Possessing Robust Oxidase-Mimetic Catalytic Function" Wang and coworkers significantly improved the quality of their work, with respect to the analysis and interpretation of the presented EPR data. My suggestions and questions have been addressed in the rebuttal letter and in the manuscript. I particularly appreciate the paramagnetic NMR data presented in the revised manuscript, which nicely complements the EPR results and adds additional important information on the coordination structure of the copper site in the prepared self-assembled enzymes. I therefore, recommend publication of this manuscript as is in Nature Communications.

Reviewer 1

The authors have addressed my concerns, and I support the acceptance of this work.

Our Response: We are very grateful to the reviewer for approving our work.

Reviewer 2

With some turnover established, it's good to see that the reported assemblies are in fact catalytic and the reported kinetic parameters are valid. I'm puzzled by the authors' response that high absorbance in their kinetic assay precludes the proper turnover determination, though. What happened to dilution and/or shorter pathway cuvettes? It's not really essential, as TN of 4 is ok.

Our Response: We thank reviewer for the affirmation. In the time of 1000 seconds, we can see that one copper can promote oxidation of ca. sixty 2,4-DCP molecules, indicating Fmoc-K/GMP/Cu²⁺ is a catalyst. We have added the description of this section to the manuscript.

The discussion has been added to Page 9 Line 18 of the revised manuscript, “As shown in Supplementary fig. 34a, in the presence of Fmoc-K/GMP/Cu²⁺ (Cu²⁺ 5 μM) and 2,4-DCP (1 mM)/4-AP (1 mM), the absorbance reached to the 4 when the catalyzed reaction proceeded for ca. 10 min. After calculating, one copper can promote oxidation of at least ca. sixty 2,4-DCP molecules, indicating Fmoc-K/GMP/Cu²⁺ is a catalyst.”

Reviewer 3

In the latest revised version of the manuscript “A Supramolecular Metalloenzyme Possessing Robust Oxidase-Mimetic Catalytic Function” Wang and coworkers significantly improved the quality of their work, with respect to the analysis and

interpretation of the presented EPR data. My suggestions and questions have been addressed in the rebuttal letter and in the manuscript. I particularly appreciate the paramagnetic NMR data presented in the revised manuscript, which nicely complements the EPR results and adds additional important information on the coordination structure of the copper site in the prepared self-assembled enzymes. I therefore, recommend publication of this manuscript as is in Nature Communications.

Our Response: We are very grateful to the reviewer for approving our work.